# Coordinated early immune response in the lungs is required for effective control of SARS-CoV-2 replication

Klara Lenart [1,2,3,7], Hendrik Feuerstein [1,2,3], Mariana Prado Marmorato[1,2,3], Laura Perez Vidakovics [4], Gerald McInerney [4], Mimi Guebre-Xabier[5], Jessica F. Trost [5], Bengt Eriksson[6], Gale Smith [5], Nita Patel[5] & Karin Loré [1,2,3] ✉

Despite waning of virus-neutralizing antibodies, protection against severe SARS-CoV-2 in the majority of immune individuals remains high, but the underlying immune mechanisms are incompletely understood. Here, rhesus macaques with pre-existing immunity from Novavax WA-1 and/or P.1 vaccines and WA-1 or P.1 infection are immunized with a bivalent WA-1/Omicron BA.5 Novavax vaccine ten months after the last exposure. The boost vaccination primarily increases the frequency of cross-reactive spike (S)-specific antibodies and B cells instead of inducing de novo BA.5-specific responses. Reinfection with heterologous Omicron XBB.1.5 six months after the boost vaccination results in low levels of virus replication in the respiratory tract compared with virus-naïve results from other studies. Whereas systemic S-specific immunity remains largely unchanged in all animals, the animals with complete protection from infection exhibit a stronger influx of S-specific IgG, monocytes, B cells and T cells into the bronchioalveolar space combined with expansion of CD69⁺CD103⁺ lung tissue-resident, S-specific CD8 T cells compared to actively infected animals. Our results underscore the importance of localized respiratory immune responses in mediating protection from Omicron reinfection and provide guidance for future vaccine development.

Immunity conferred by infection or widespread vaccination jointly reduced the impact of the SARS-CoV-2 pandemic. The extent of infections with the Delta and Omicron variants led to most individuals today having a combination of infection- and vaccine-derived immunity, described as hybrid immunity[1,2]. Prior infection enhances the vaccine response by increasing the potency and breadth of systemic antibodies and augmenting mucosal immunity, particularly virus-specific IgA responses[3–7]. In epidemiological studies, hybrid immunity also provided increased protection against infection with Omicron variants compared to vaccination or prior infection alone[8–13].

However, observational studies of SARS-CoV-2 hybrid immunity in humans are increasingly difficult due to the high diversity of past exposures through vaccination and infection with different variants. Abortive and asymptomatic infections, when the individuals are unaware of their complete infection history, add another layer of complexity[14]. On the other hand, controlled SARS-CoV-2 exposures in

¹Division of Immunology and Respiratory Medicine, Department of Medicine Solna, Karolinska Institutet, Stockholm, Sweden. ²Karolinska University Hospital, Stockholm, Sweden. ³Center for Molecular Medicine, Karolinska Institutet, Stockholm, Sweden. ⁴Division of Virology and Immunology, Department of Microbiology, Tumor and Cell Biology, Karolinska Institutet, Stockholm, Sweden. ⁵Novavax Inc, Gaithersburg, MD, USA. ⁶Astrid Fagraeus Laboratory, Comparative Medicine, Karolinska Institutet, Stockholm, Sweden. ⁷Present address: Laboratory of Molecular Immunology, Rockefeller University, New York, NY, USA. ✉e-mail: karin.lore@ki.se

non-human primate (NHP) studies provide an opportunity to dissect the responses at mucosal sites in addition to the periphery for a comprehensive view of the elicited immunity.

Emergence of SARS-CoV-2 Omicron in November 2021 raised concerns about viral escape with >30 mutations in the Spike (S) protein[15] and considerable loss of neutralizing activity in vaccinated individuals[16]. This prompted an update of existing vaccines to better match circulating variants, leading to the licensing of WA-1/BA.5 bivalent booster mRNA vaccines in the fall of 2022. Meanwhile, an XBB Omicron lineage started to prevail in Europe and North America. This lineage arose through recombination of BA.2.10.1 and BA.2.75 and has shown one of the highest immune evasion rates to date[17–19]. Nevertheless, the risk of severe outcomes was reduced during the post-Omicron era compared to the earlier variants, owing to an intrinsic reduction in disease severity as well as substantial pre-existing immunity in the population[20,21].

In uninfected vaccinated individuals, neutralizing antibodies in the serum are highly associated with protection from infection[22–24]. However, the rapid viral evolution led to a significant reduction in neutralization potency[17–19], and broadly neutralizing antibodies are rarely elicited in protective quantities[25,26]. The specificity of B cell responses has been suggested to be imprinted by the primary exposure to the antigen, and immunization with licensed variant vaccines predominantly leads to the expansion of cross-reactive memory B cells with limited formation of de novo responses against variant-specific epitopes[27–31]. Elicitation of highly cross-reactive and broadly neutralizing memory B cells is thus critical for establishing antibody responses capable of neutralizing current and future SARS-CoV-2 lineages. Conversely, T cell responses show a high level of cross-reactivity between ancestral and Omicron-derived S antigens[32,33], and CD8 T cells were shown to contribute to a reduction in viral loads after challenge in animal models[24,34,35].

Robust mucosal immunity is likely key for effective control of viral replication at the site of infection. Respiratory infections such as influenza mobilize the recruitment of mononuclear phagocytes into the respiratory tract[36,37] and induce tissue-resident memory cells, which are primed through local antigen encounter at the site of infection and poised to react when the antigen is re-encountered[38–41]. While tissue-resident memory cells can be established through natural infection[38–41] or mucosal vaccination[39,42–45], it remains unclear whether intramuscular immunization has the ability to boost pre-existing mucosal immune responses.

Here, a group of rhesus macaques with unique pre-existing hybrid immunity receive a bivalent WA-1/BA.5 S protein booster vaccine, followed by an Omicron XBB.1.5 infection. This model allows for an in-depth study of the evolution and maintenance of the S-specific adaptive immune responses across tissues and SARS-CoV-2 exposures, enabling a dissection of antibody, B cell and T cell memory. We find that lung infiltration of immune cells, including monocytes and tissue-resident memory T cells is a key feature of effective recall immune responses upon SARS-CoV-2 reinfection, providing insights for the development of future vaccines, immunization schedules and routes.

## Results

### Bivalent COVID-19 immunization of hybrid immune rhesus macaques

Hybrid immune NHPs were generated through immunization with the clinical dose (5 μg) of Novavax's approved protein vaccine NVX-CoV2373 based on WA-1 and/or the updated NVX-CoV2443 based on P.1 (Gamma variant) followed by a high dose viral challenge approximately 30 weeks (7 months) later (Fig. 1A)[46]. Details about each NHPs exposure history are listed in Supplementary Table 1. While immunized animals controlled the infection significantly better than naïve unimmunized animals, all animals cleared the virus within 14 days and remained closely monitored for waning immunity (Supplementary

Fig. 1A). At study week 108 (10 months after the first infection), 12 hybrid-immune animals received a bivalent WA-1/BA.5 booster vaccine, composed of 2.5 μg NVX-CoV2373 (WA-1) and 2.5 μg NVX-CoV2540 (BA.5) S proteins. Three macaques that were never immunized but received the earlier viral challenge along with the rest of the animals were designated as the non-immunized comparator group (Fig. 1A, Supplementary Fig. 1A).

After the bivalent booster, WA-1 and BA.5 S- and receptor binding domain (RBD)-specific IgG antibody titers in hybrid immune NHPs increased at least 10-fold and remained above pre-boost levels for 27 weeks, while the responses in non-immunized infected-only group were stable during the follow-up period (Supplementary Fig. 2A–E). In line with this data, a strong increase in neutralizing antibody responses against both WA-1 and BA.4/5 was observed after the bivalent booster immunization in hybrid immune NHPs, with a more prominent rise in BA.4/5 neutralization (Fig. 1B–D). In addition, we measured neutralizing antibodies against different Omicron lineages before and after the bivalent booster and observed a strong increase in neutralization of BA.1, BA.2, BA.5 and BQ.1.1 as well as a moderate increase in XBB.1.5 neutralization (Fig. 1E, F). Four weeks after the boost immunization, S-specific plasma cells in the bone marrow were readily detectable in all hybrid-immune animals, whereas no or low numbers were found in the infected-only group (Supplementary Fig. 2F).

To further dissect the specificity of the B cell response, we evaluated the IgG+ B cells using a dual WA-1 and BA.5 S probing strategy to enumerate the WA-1+ and BA.5+ single-positive versus WA-1/BA.5+ cross-reactive memory B cells (Fig. 1G, Supplementary Fig. 2G). Degree of non-specific binding was evaluated using a PBMC sample from a SARS-CoV-2 naïve NHP (Supplementary Fig. 2H). In response to the bivalent boost, S-reactive memory B cells expanded in hybrid immune animals (Fig. 1H). In particular, we observed an increase in the frequency of WA-1+ and WA-1/BA.5+ B cells, but not BA.5+ B cells, compared to the baseline levels (Fig. 1I, J). Together, these data indicate that bivalent vaccination can boost Omicron-specific immunity predominantly through expansion of cross-reactive B cell clones.

Repeated exposure to the antigen has been shown to enhance B cell responses[5,47]. Indeed, antibody and memory B cell responses in the group with five exposures (blue) were elevated compared to the group with four exposures (red), and significantly higher compared to the infected-only comparator group (Supplementary Fig. 2I–K). This indicates that repeated antigen dosing leads to accumulation of specific B cell responses.

### Omicron XBB.1.5 infection is mild in animals with pre-existing immunity

At week 27 (6 months after the bivalent boost), all animals received a viral challenge with the Omicron XBB.1.5 strain (Fig. 1A). The extent of XBB.1.5 infection in the lower and upper respiratory tract was evaluated by RT-qPCR assay[27,46], quantifying genomic and subgenomic N gene transcripts (gN and sgN, respectively). sgN transcripts, indicating the presence of replicating virus, were detected in seven (lower airways) and ten (upper airways) out of 15 NHPs at days 2–7, but all animals were sgN negative 14 days after infection (Fig. 2A). Peak sgN titers in the bronchoalveolar lavage (BAL) samples measured up to $10^5$ copies per sample (Fig. 2B). Although our study did not include naïve rhesus macaques for comparison, earlier Omicron challenge experiments with naive NHPs that used identical method and dose of viral inoculation as well as sgRNA detection, showed viral loads in the respiratory tract in the range of $10^5$–$10^8$ copies per sample[27,45,48], which is several logs higher than in the pre-immune animals in our study.

Some level of XBB.1.5 replication was detected in 12 out of 15 animals (Fig. 2C). Although the study was not powered to assess vaccine efficacy after vaccination, there were no significant differences in peak (Fig. 2B) or area under curve (AUC) viral load (Supplementary Fig. 3A). Moreover, one NHP in the infected-only group was not

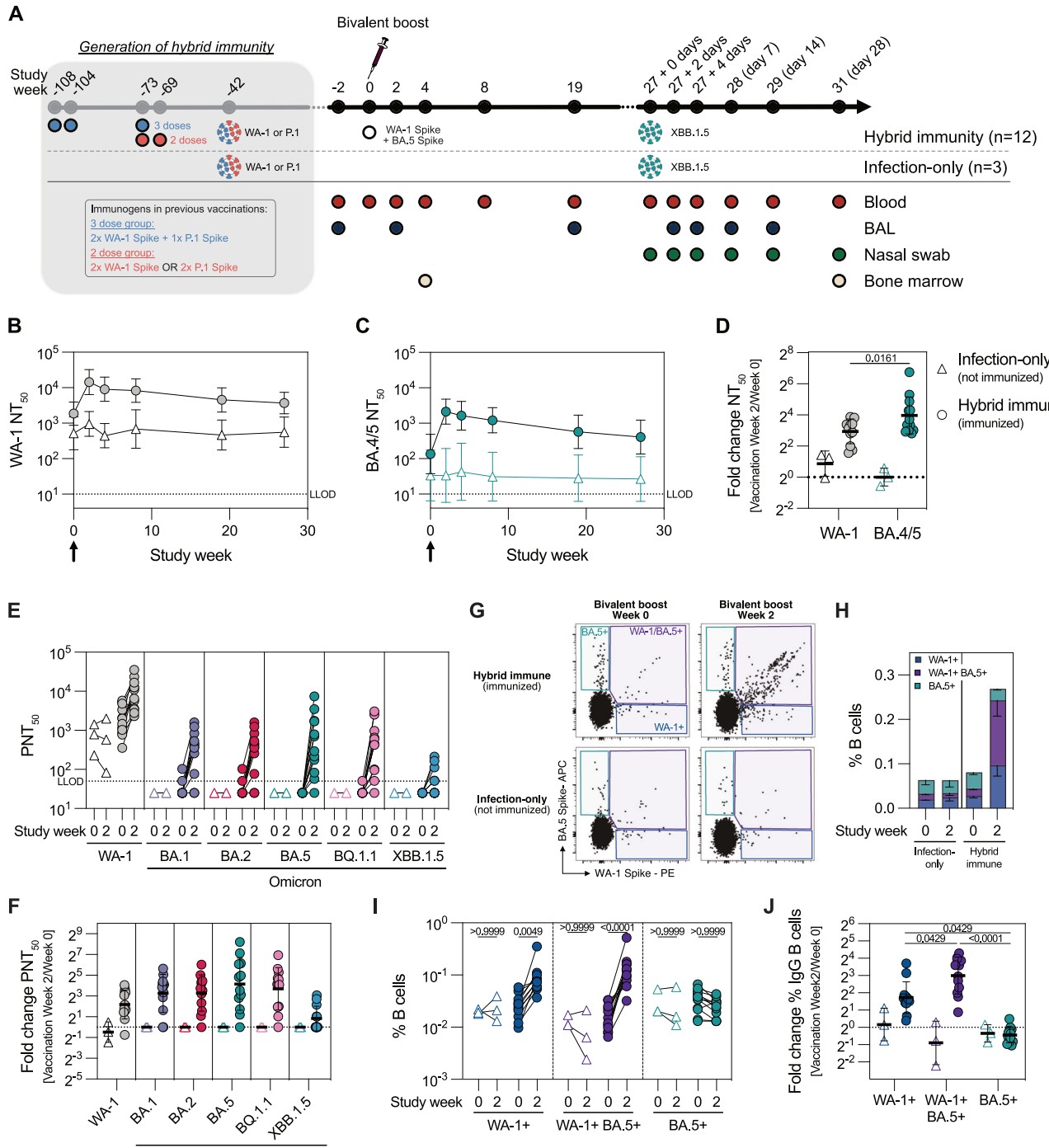

**Fig. 1 | WA-1/BA.5 bivalent booster immunization of hybrid immune animals expands cross-reactive B cell responses. A** Study design, indicating immunization and infection history, and blood and tissue sampling schedule. Hybrid immune and infected-only rhesus macaques were generated in an earlier study (gray-shaded rectangle) through protein subunit immunization (hybrid immune) and infection with SARS-CoV-2 WA-1 or P.1 strains (hybrid immune and infected-only). **B, C** Serum neutralizing titers after bivalent booster immunization against SARS-CoV-2 WA-1 (**B**) and Omicron BA.4/5 (**C**). Arrow indicates immunization (n = 3-12). Infected-only group was not immunized. **D** Fold increase in WA-1 and BA.4/5 neutralizing titers two weeks after immunization compared to pre-immunization titers (n = 3–12). Statistical analysis was performed only using the data from hybrid immune NHPs. **E, F** Neutralization of WA-1 and different Omicron sublineages before and after bivalent booster immunization (**E**) and the fold increase in neutralizing titers two weeks after immunization compared to pre-immunization titers (**F**) (n = 3–12).

**G** Representative flow cytometry plots depicting Spike-specific IgG memory B cells before and after bivalent booster. **H** Cumulative frequency of Spike-specific IgG memory B cells in the blood before and after bivalent booster (n = 3–12).
**I** Frequencies of IgG memory B cells of different specificities in the blood before and after bivalent booster (n = 3-12). **J** Fold increase in IgG memory B cells of different specificities two weeks after immunization compared to pre-immunization (n = 3-12). Statistical analysis was performed only using the data from hybrid immune NHPs. Data is presented as geometric mean ± geometric SD (**B–D**, **F**, **J**) or arithmetic mean ± SEM (**H**). Horizontal dotted line represents a lower level of detection (**B**, **D**, **E**) or a fold change of 1 (**D**, **F**, **J**). Statistical analysis was performed using a non-parametric two-tailed Wilcoxon test (**D**), non-parametric Kruskal-Wallis test with Dunn's correction for multiple analysis (**I**) or Friedman test with Dunn's correction for multiple analysis (**J**), LLOD lower level of detection. Source data are provided as a Source Data file.

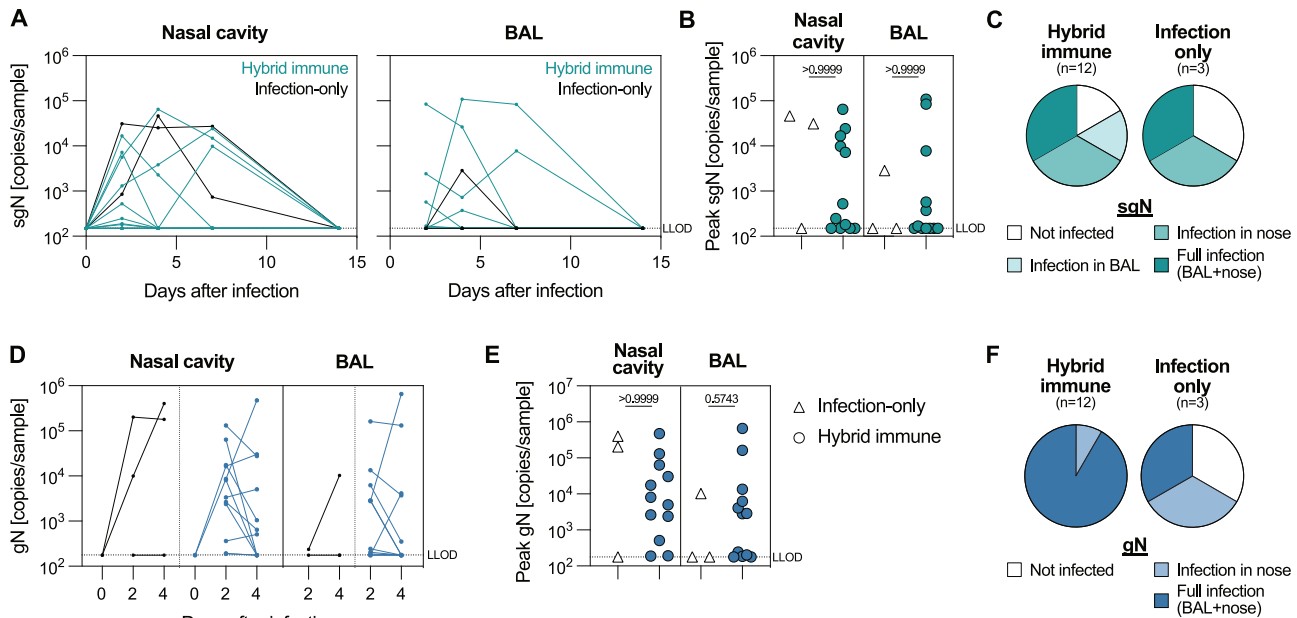

**Fig. 2 | Limited viral replication in the respiratory tract of pre-immune animals after XBB.1.5 infection. A** Active viral replication of SARS-CoV-2 XBB.1.5 virus in the nasal cavity and lungs (samples collected by nasal swabs and BAL, respectively) assessed by subgenomic (sg)N gene-targeted RT-qPCR ($n = 3$–12 biological replicates per group). **B** Peak sgN copies in the upper and lower airways ($n = 3$–12 biological replicates per group). **C** Proportion of animals in each group with detectable sgN transcripts in the nasal cavity and/or lungs, or lack thereof, at any point after infection ($n = 3$–12 biological replicates per group). **D** Presence of SARS-CoV-2 XBB.1.5 virus in the nasal cavity and lungs (samples collected by nasal swabs and BAL, respectively) assessed by genomic (g)N gene-targeted RT-qPCR ($n = 3$–12 biological replicates per group). **E** Peak gN copies in the upper and lower airways ($n = 3$–12 biological replicates per group). **F** Proportion of animals in each group with detectable gN transcripts in the nasal cavity and/or lungs, or lack thereof, at any point after infection ($n = 3$–12 biological replicates per group). Horizontal dotted line represents a lower level of detection (**A, B, D, E**). Statistical analysis was performed using a non-parametric Kruskal-Wallis test with Dunn's correction for multiple analysis (**B, E**). LLOD lower level of detection. Source data are provided as a Source Data file.

sampled on day 2 due to a small wound in the trachea, possibly leading to lower observed peak viral loads for this animal.

In addition to sgN, gN transcripts were quantified at days 2–4 after infection, and the majority (93%) of the animals had detectable gN transcripts in the respiratory tract on days 2–4 after infection (Fig. 2D–F). This suggested that the virus was present but unable to replicate at detectable levels. sgN and gN showed a linear relationship at higher viral loads, but sgN was undetectable when gN copies were below $10^3$ copies/sample (Supplementary Fig. 3B). It has been reported that isolation of live virus after Omicron challenge is only possible from samples with high number of sgN transcripts[45,48,49]. Taken together, this suggests a high level of protection against infection with XBB.1.5 in animals with pre-existing immunity.

A noticeable feature of Omicron infection compared to pre-Omicron variants is altered tropism favoring the upper respiratory tract instead of the lungs[50–52]. Indeed, we observed that an active XBB.1.5 infection was more prevalent in the nasal compartment compared to the lungs (Fig. 2C, F), but there was no difference in infection rates between the hybrid immune and infection-only NHPs (Fig. 2B, C, E, F), although statistical comparison is difficult due to uneven group sizes.

Successful RNA extraction was confirmed by detection of RNAseP transcripts, a ubiquitously expressed ribozyme, where only 6 nasal swab samples (out of 75, 8.0 %) had Ct values above 39 (Supplementary Fig. 3C). Moreover, viral transcripts were detectable even in nasal swab specimens with high RNAseP Ct values (Supplementary Fig. 3D), confirming high sensitivity of the sgN and gN RT-qPCR assays.

Finally, we found increased IgG and T cell infiltration in the BAL after infection compared to pre-infection timepoints (Supplementary Fig. 3E–G), demonstrating that virus-induced lung inflammation recruits immune cells to the site of infection. Together, these findings show that XBB.1.5 infection in animals with pre-existing immunity is mild, as most animals exhibited limited or fully suppressed viral replication early in the infection. Nevertheless, infiltration of IgG antibodies and T cells to the site of infection clearly suggested that the infection induced local immune activation.

## Limited boosting of peripheral immune responses upon XBB.1.5 infection in animals with pre-existing immunity

B cell responses after XBB.1.5 infection were evaluated using the same methods as after the bivalent boost immunization. In hybrid immune animals, WA-1 and BA.5 S-specific plasma IgG antibodies did not increase after XBB.1.5 infection. Conversely, a clear expansion was detected in all infected-only animals and both groups displayed similar plasma IgG titers 14-28 days after infection (Supplementary Fig. 4A, B). Neutralizing titers against WA-1 remained stable at all post-infection timepoints in both groups, whereas neutralizing titers against BA.4/5 increased primarily in the infected-only group (Fig. 3A, B). We also measured neutralizing antibody titers against a panel of Omicron sublineages using a different pseudoneutralization assay, which agreed with our data that while XBB.1.5 infection elevates neutralizing titers against Omicron variants approximately 2-fold, WA-1 titers remain unchanged (Fig. 3C, D). Although the overall increase in antibody responses after immunization was superior to the increase after infection, these data suggest that even mild infection can boost immunity in individuals with low pre-existing responses.

In line with antibody titers, S-specific memory B cells expanded in both the hybrid immune and infected-only group after XBB.1.5 infection (Fig. 3E–G), albeit to a lower degree than observed after bivalent immunization (Fig. 1J). The highest fold change was again detected for the cross-reactive WA-1/BA.5+ B cells in both groups, with no change in the BA.5+ only population (Fig. 3H).

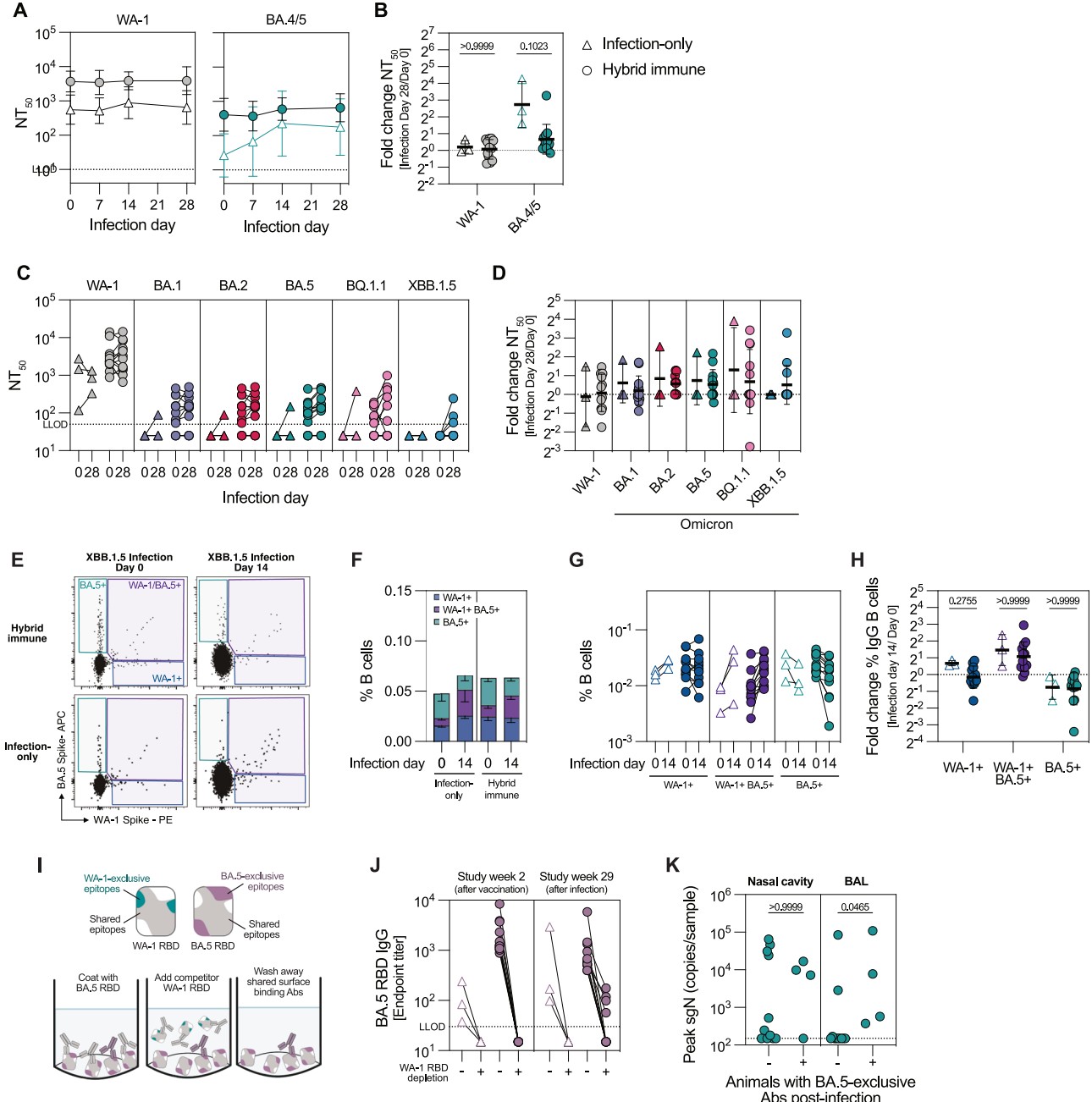

**Fig. 3 | Peripheral B cell responses to XBB.1.5 infection. A** Serum neutralizing titers after XBB.1.5 infection against SARS-CoV-2 WA-1 and Omicron BA.4/5 (n = 3–12 biological replicates per group). **B** Fold increase in neutralization titers four weeks after infection compared to pre-infection titers (n = 3–12 biological replicates per group). **C, D** Neutralization of WA-1 and different Omicron sublineages before and after XBB.1.5 infection (**C**) and the fold increase in neutralizing titers four weeks after infection compared to pre-infection titers (**D**) (n = 3–12). **E** Representative flow cytometry plots depicting WA-1 and BA.5 S specificity of the IgG memory B cells before and after XBB.1.5 infection. **F** Cumulative frequency of Spike-specific IgG memory B cells in the blood before and after XBB.1.5 infection (n = 3–12). **G** Frequencies of IgG memory B cells of different specificities in the blood before and after XBB.1.5 infection (n = 3–12). **H** Fold increase in IgG memory B cells of different specificities two weeks after XBB.1.5 infection compared to pre-infection

(n = 3–12 biological replicates per group). **I** Schematic representation of competition ELISA assay to evaluate the proportion of antibodies binding epitopes present on BA.5 but not WA-1 RBD. **J** Binding of plasma IgG antibodies to BA.5 RBD with and without depletion of WA-1 RBD-specific antibodies, two weeks after bivalent booster and XBB.1.5 infection (n = 3–12 biological replicates per group). **K** Peak sgN viral loads in the nasal cavity and lungs of animals with no evidence of de novo response against BA.5 RBD (n = 11) and the ones with remaining detectable BA.5 binding after depletion with WA-1 RBD in post-infection samples (n = 4). Data is presented as geometric mean ± geometric SD (**A, B, D, H**) or arithmetic mean ± SEM (**F**). Horizontal dotted line represents a lower level of detection (**A, C, J, K**) or a fold change of 1 (**B, D, H**). Statistical analysis was performed using a non-parametric Kruskal-Wallis test with Dunn's correction for multiple analysis (**B, H, K**). LLOD lower level of detection. Source data are provided as a Source Data file.

This prompted us to evaluate the proportion of IgG plasma antibodies binding epitopes unique to BA.5 or WA-1 RBD, using a competition ELISA assay where RBD-specific IgG antibodies binding shared surface area are depleted (Fig. 3I). As reported in clinical studies with

bivalent mRNA vaccine boosters[30], no BA.5 RBD-binding was detected in WA-1 RBD-depleted samples collected after immunization (Fig. 3J), whereas a large proportion of WA-1 RBD IgG remains bound even after depletion with BA.5 RBD (Supplementary Fig. 4C). After XBB.1.5

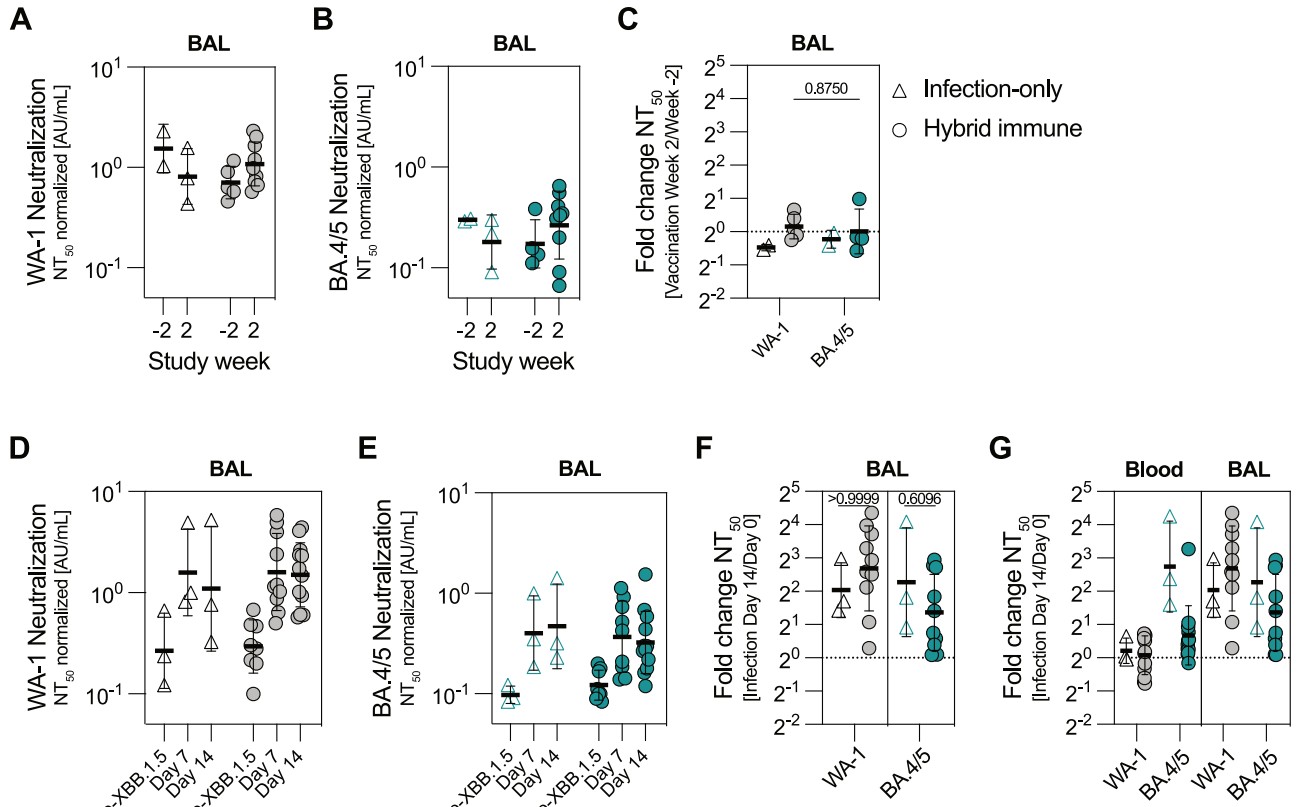

**Fig. 4 | Mucosal antibody responses after bivalent immunization and XBB.1.5 infection. A, B** Neutralizing antibody titers against SARS-CoV-2 WA-1 (**A**) and Omicron BA.4/5 (**B**) in the BAL fluid after immunization ($n$ = 2–9 biological replicates per group). **C** Fold increase in neutralizing titers in the BAL fluid two weeks after bivalent immunization compared to pre-immunization timepoint ($n$ = 2–4 biological replicates per group). **D, E** Neutralizing antibody titers against SARS-CoV-2 WA-1 (**D**) and Omicron BA.4/5 (**E**) in the BAL fluid after XBB.1.5 infection ($n$ = 3–10 biological replicates per group). **F** Fold increase in neutralizing titers in the BAL fluid two weeks after XBB.1.5 infection compared to pre-infection timepoint ($n$ = 3–10 biological replicates per group). **G** Comparison of fold increase in

WA-1 and BA.4/5 neutralization in the blood and BAL fluid after XBB.1.5 infection ($n$ = 3–12 biological replicates per group). Related to Fig. 3B, 4F. All endpoint and neutralizing titers in the BAL are normalized to the retrieved volume of the BAL fluid. Samples with total IgG endpoint titer <1000 (Supplementary Fig. 3D) were excluded from the analysis. Data is presented as geometric mean ± geometric SD (**A–G**). Horizontal dotted line represents a fold change of 1 (**C, F, G**). Statistical analysis was performed using a non-parametric two-tailed Wilcoxon test (**C**) or a non-parametric Kruskal-Wallis test with Dunn's correction for multiple analysis (**F**). Source data are provided as a Source Data file.

infection, however, IgG antibodies binding BA.5 RBD-exclusive epitopes were detected in four animals, which all showed evidence of active infection in the BAL (Fig. 3K), suggesting that a sufficient antigen load is required for elicitation of broad responses. Overall, around 50% of IgG serum antibodies preferentially bound epitopes unique to WA-1 RBD after bivalent vaccination and XBB.1.5 infection (Supplementary Fig. 4D). This data suggests that the IgG serum antibody reactivity was heavily imprinted by the viral strain from the primary exposure, and that de novo BA.5-specific B cell response was rare in response to bivalent booster immunization but could be elicited after Omicron infection.

**Increased mucosal responses after immunization and infection**

Robust immunity at mucosal sites, in the form of humoral and cellular memory, can aid in rapid neutralization of pathogens. In NHPs, we can access the lower airways through longitudinal non-invasive BAL sampling and thus evaluate both antibody and cellular immunity in the mucosa. To ensure high accuracy of our results, only BAL samples with sufficient total IgG content (endpoint titer > 1000) were used in the following analyses and all antibody titers were normalized to the total volume of retrieved BAL fluid.

In line with plasma IgG, WA-1 and BA.5 S-specific antibodies in the BAL fluid increased considerably two weeks after bivalent boost in the hybrid immune group (Supplementary Fig. 5A–C). Nevertheless, we

did not detect increased neutralization titers in the BAL fluid at the same timepoint (Fig. 4A–C) despite the substantial increase in the serum (Fig. 1D). Two days after XBB.1.5 infection, a rapid anamnestic antibody response was detected in the BAL fluid, which remained elevated throughout the follow-up period of 14 days in both groups (Supplementary Fig. 5D, E). However, the fold increase in IgG antibodies after infection was higher in the infected-only group both in BAL and blood (Supplementary Fig. 5F, Fig. 3B). Similarly, WA-1 and BA.4/5 neutralizing titers were increased in the BAL fluid of all animals after XBB.1.5 infection (Fig. 4D–F). Even though most animals showed a modest increase in WA-1 neutralization in the blood after infection, both WA-1 and BA.4/5 neutralizing titers in the BAL were boosted to a similar degree (Fig. 4G). Antibodies in the lower respiratory tract may therefore be partly produced by locally reactivated plasma cells of distinct specificity compared to the systemic B cell compartment.

Several studies have demonstrated a high degree of conservation in T cell epitopes across SARS-CoV-2 variants[32,33,53]. A recall assay using an overlapping peptide library derived from WA-1 S (Supplementary Fig. 6A, B) showed considerable expansion of S-specific Th1 CD4 T cells and circulating T follicular cells (cTfh) in the blood two weeks after bivalent booster (Fig. 5A, Supplementary Fig. 6E), with minor increases in S-specific Th2, Th17 and CD8 T cells (Supplementary Fig. 6C, D, Fig. 5B). In contrast, two weeks after infection, we detected an increase in S-specific Th1 and Th17 memory T cells, as well as a notable

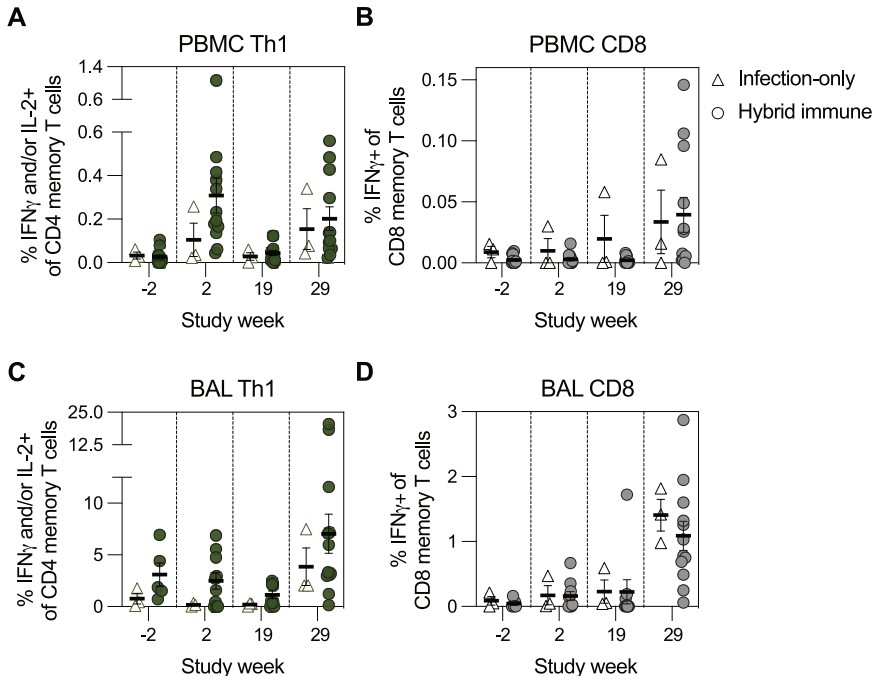

**Fig. 5 | T cell immunity after bivalent immunization and XBB.1.5 infection. A, B** S-specific Th1 memory T cells in the blood (**A**) and BAL (**B**) before and after bivalent immunization and XBB.1.5 infection (*n* = 3–12 biological replicates per group). **C, D** S-specific CD8 memory T cells in the blood (**C**) and BAL (**D**) before and after bivalent immunization and XBB.1.5 infection (*n* = 3–12 biological replicates per group). Data is presented as arithmetic mean ± SEM (**A**–**D**). Source data are provided as a Source Data file.

expansion of CD8 T cells (Fig. 5A, B, Supplementary Fig. 4D). cTfh cells did not detectably expand in response to XBB.1.5 challenge, in line with the lack of significant B cell expansion after infection (Supplementary Fig. 6E, Fig. 3H, I). While boost immunization did not have a significant influence on the frequency of S-specific memory T cells In the lungs, XBB.1.5 infection clearly expanded both S-specific CD4 and CD8 memory T cells (Fig. 5C, D, Supplementary Fig. 6F–H). Together, these results show that a rapid anamnestic T cell and antibody response to Omicron XBB.1.5 infection is restricted to the respiratory tract.

### Coordinated early immune response in the respiratory tract after SARS-CoV-2 infection

As mentioned previously, eight of the 15 NHPs had undetectable sgN copies in the BAL fluid after XBB.1.5 infection, suggesting that the viral replication was rapidly suppressed in this group, granting a significant level of protection from infection (Fig. 6A). As the peripheral immune responses were minimally perturbed in response to XBB.1.5, we focused on the immune responses in the lungs at 2–14 days after infection to investigate the parameters separating the protected (PCR-negative) animals from actively infected (PCR-positive) ones.

Protected animals had moderately increased total and BA.5 S-specific IgG antibodies 4 days after XBB.1.5 infection compared to infected NHPs (Fig. 6B, C), with no differences at other timepoints. This suggested that early immune events may dictate the course of infection and led us to analyze the composition of the cellular BAL fraction two and four days after infection (Supplementary Fig. 7A). Due to limited BAL sample availability, the analysis was focused on the bulk composition of infiltrating immune cells rather than detailed profiling of antigen specificity of BAL-infiltrating lymphocytes. In protected animals, we observed a significant influx of immune cells, particularly B cell and monocyte subsets, into the BAL four days after infection compared to steady state (Fig. 6D, E). While there are few B cells and monocytes present in the BAL at steady state, we found a consistent increase in frequency of these subsets in the BAL across all protected

animals after infection, while the extent of immune infiltration in the BAL of infected NHPs was more varied and often low (Fig. 6F). Notably, four days after XBB.1.5 infection, the frequency of T cells and monocytes in the BAL of protected group was significantly higher compared to the infected animals (Fig. 6G).

Obtaining samples from the respiratory tract for assessment of protection from infection is difficult, and therefore serum antibody titers have been proposed as a correlate of protection for SARS-CoV-2 infection[23,24,54]. However, the correlates were mainly assessed in studies where immunization and infection strains were homologous. In a highly divergent setup like the one described here, higher pre-infection serum neutralizing titers against Omicron BA.2, a variant closely related to XBB.1.5, did not correlate with reduced viral loads (Supplementary Fig. 7B). Conversely, BA.5 S-specific IgG as well as the frequency of CD69 + CD103 + CD8 cells, also known as tissue-resident memory T ($T_{RM}$) cells in the BAL four days after infection were associated with reduced viral load after challenge (Fig. 6H, I). The increase in BA.5 S IgG titers also correlated with elevated CD8 $T_{RM}$ cell frequency (Fig. 6J). On the other hand, the increased BA.5 IgG BAL antibodies did not correlate with increased monocyte frequency in the BAL after Omicron reinfection (Fig. 6K), even though higher monocyte frequency was associated with reduced viral loads as well as increased frequency of CD8 $T_{RM}$ cells (Fig. 6L, M). This supports the notion that rapid and robust innate and adaptive responses in the respiratory tract are critical for effective control of viral replication.

### Expansion of tissue-resident memory T cells after reinfection in protected animals

$T_{RM}$ cells, generated in response to site-specific antigen encounter and poised to mediate rapid in situ immune responses, have been associated with protective effects against infectious diseases[55], but are poorly induced by intramuscular immunization[56]. Two weeks after the XBB.1.5 infection, we detected a population of S-specific CD8 T cells in the BAL that co-expressed the canonical $T_{RM}$ markers CD69 and CD103

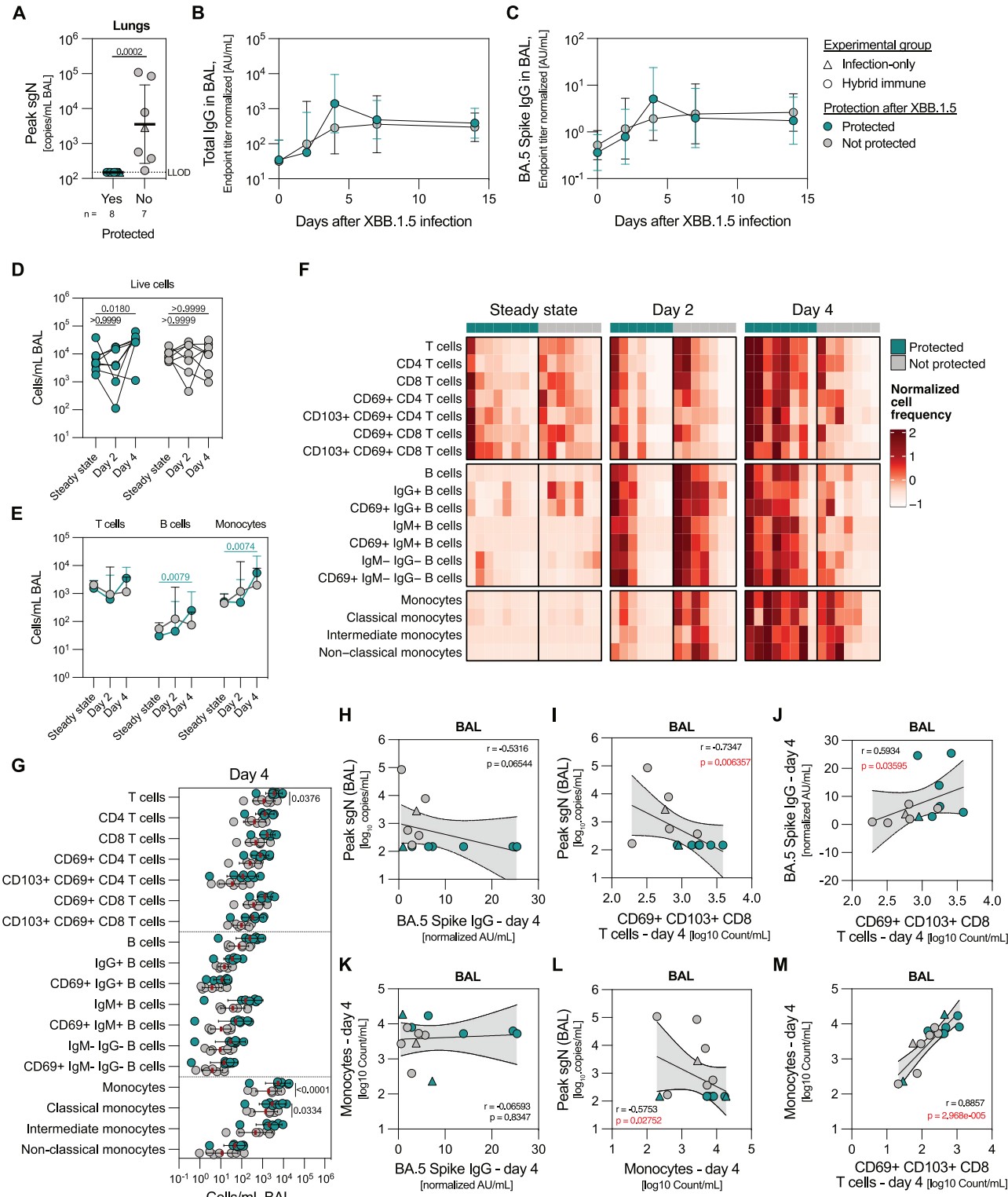

(Fig. 7A). S-specific $T_{RM}$ cells also downregulated CCR7 expression compared to the naïve CD8 T cells in blood (Fig. 7B), corroborating their tissue-resident phenotype. While S-specific CD8 T cells in the blood were predominantly effector memory T cells ($T_{EM}$) with minor proportions of central memory ($T_{CM}$) and terminally differentiated effector cells ($T_{EMRA}$), all CD8 T cells in the BAL, including S-specific CD8 T and $T_{RM}$ cells, were of $T_{EM}$ phenotype (Fig. 7C, D).

The immune response to the S protein in these animals was comprehensively assessed across multiple immunizations and infections, which prompted us to compare the $T_{RM}$ responses after the primary immunization and WA-1/P.1 infection to the responses elicited by the bivalent booster and later the XBB.1.5 infection (Fig. 7E). While intramuscular immunizations did not expand the S-specific CD8 T cells in the blood nor BAL, increased frequencies were observed after both SARS-CoV-2 infections (Fig. 7F–H). Two weeks after XBB.1.5 infection, protected NHPs had lower frequencies of S-specific CD8 memory T cells in the blood compared to the infected group, but exhibited a marked expansion of S-specific CD8 and $T_{RM}$ populations in the BAL (Fig. 7I–K). The elevated S-specific T cell response in the BAL was restricted to the CD8 T cell compartment, demonstrated by a lack of

**Fig. 6 | Early infiltration of antibodies and immune cells into mucosal sites is critical for full control of viral replication. A** Peak sgN viral loads in the lungs separate the animals into protected (i.e. no viral replication) and not protected (i.e. detectable infection) group (n = 7–8 biological replicates per group). **B, C** Concentration of total IgG (**B**) and BA.5 S-specific IgG (**C**) antibodies in the BAL fluid after XBB.1.5 infection (n = 7–8 biological replicates per group) in protected and not protected groups. **D, E** Normalized frequencies of live immune cells (**D**) and different immune cell subsets (**E**) in the BAL at steady state, two and four days after infection (n = 7–8 biological replicates per group). **F** Normalized frequencies of immune cell populations in the BAL at steady state, two and four days after infection (n = 7–8 biological replicates per group). **G** Normalized frequencies of immune cell populations in the BAL four days after infection (n = 7–8 biological replicates per group). **H, I** Correlation between peak sgN viral loads in BAL and concentration of BA.5 S-specific IgG in the BAL at day 4 (**H**) and concentration of CD8 $T_{RM}$ cells in

the BAL at day 4 (**I**). **J** Correlation between concentration of CD8 $T_{RM}$ cells and BA.5 S-specific IgG in the BAL four days after XBB.1.5 infection. **K–M** Correlation between monocyte frequency and BA.5 S-specific IgG (**K**), peak sgN loads (**L**) and frequency of CD8 TRM cells (**M**) in the BAL at day 4 after XBB.1.5 infection. All endpoint titers and cell counts in the BAL are normalized to the retrieved volume of the BAL fluid. Data is presented as geometric mean ± geometric SD (**A–C, E, G**). Horizontal dotted line represents a lower level of detection (**A**). Statistical analysis was performed using a non-parametric two-tailed Mann-Whitney test (**A**), non-parametric Kruskal-Wallis test with Dunn's correction for multiple analysis (**D, E, G**) or two-tailed Spearman correlation (**H–M**). The correlation plots contain a simple linear regression line with the shaded area as a 95% confidence interval (**H–M**). LLOD lower level of detection, CM classical monocytes, IM intermediate monocytes, NCM non-classical monocytes. Source data are provided as a Source Data file.

significant expansion of other T cell subsets after reinfection (Supplementary Fig. 8A–J). About 30 weeks after XBB.1.5 infection, we measured the frequency of S-specific CD8 $T_{RM}$ cells in the blood and BAL to evaluate the persistence of the $T_{RM}$ compartment (Supplementary Fig. 8K–M). At this timepoint, we detected similar $T_{RM}$ frequencies as before the infection, which is a significant contraction compared to the peak response at week 29. Altogether, our data strongly suggests that the first infection primed the S-specific lung-resident CD8 T cell memory population, which was able to efficiently expand and respond to reinfection.

Using the collected data, we performed a sparse partial least squares regression discriminant analysis (sPLS-DA), with the aim to identify key immune features associated with protection from reinfection. 102 variables, comprising both cellular and humoral immunity across multiple timepoints were included in the analysis. In two-dimensional space, the sPLS-DA analysis showed a distinct clustering of the protected and infected group (Fig. 7L). The variables that contributed most to the observed separation were predominantly collected after the infection and were not restricted to a specific cell subset (Fig. 7M). A separate sPLS-DA analysis focusing on specific study timepoints indicated that the S-specific B cell response after the bivalent booster may have a significant influence on the separation of the groups (Supplementary Fig. 9A, B), although the clustering was less robust compared to the combined analysis. The sPLS-DA clustering using only pre- or post-infection data was in agreement with the combined analysis (Supplementary Fig. 9C–F). While our dataset was not able to predict the protection from infection based on the pre-infection immune status alone, it clearly indicates that a well-coordinated early immune response comprised of virus-specific antibodies in the lungs, considerable expansion of pre-existing CD8 $T_{RM}$ cells as well as monocyte infiltration is required for effective control of viral replication.

## Discussion

The disease burden of SARS-CoV-2 has decreased over time despite the continuous spread of the virus and its variants[57]. While this has been attributed to a substantial degree of pre-existing immunity in the population[58–60], combined with the decreased pathogenicity of the Omicron variants[61,62], the precise immune mechanisms that modulate disease severity in pre-immune individuals have yet to be determined. In this study, we generated a set of rhesus macaques that received multiple vaccinations as well as two infections, creating a unique animal model to study hybrid immunity and immunological mechanisms of protection. Even though the group sizes in this study were small and uneven due to the scarcity of available research animals, thereby limiting statistical analysis, we found that the heterologous infection with XBB.1.5 variant in SARS-CoV-2-experienced rhesus macaques was mild and rapidly controlled, characterized primarily by expansion of mucosal S-specific immunity. Complete suppression of viral replication in the lungs correlated with an early rise in virus-specific antibody

titers, rapid increase in immune cell frequency in the BAL and expansion of tissue-resident memory T cells, providing evidence that induction of robust mucosal responses can aid in the effective control of SARS-CoV-2 reinfections.

Despite 10-fold higher serum antibody titers in the hybrid immune animals at the time of XBB.1.5 infection, we found no difference between the viral loads in the respiratory tract of the hybrid immune and infected-only groups, suggesting that even low levels of pre-existing immunity can provide protection from infection in an NHP model. The immune systems of rhesus macaques and humans are highly similar, leading to convergent responses to SARS-CoV-2 infection both in terms of tropism[27,50] and responding immune cells[63,64]. Unlike humans, it has been shown that SARS-CoV-2 infection in NHPs is predominantly mild[65] and immune responses may differ from the highly dysregulated immunity observed in severe COVID-19[66,67]. Nevertheless, the findings in NHP studies provide valuable insights into the mechanisms driving systemic and tissue-specific immunity after SARS-CoV-2 vaccination and infection.

Elicitation of antibody responses towards novel epitopes generated through antigenic drift or shift has been at the forefront of vaccine development efforts with both influenza and SARS-CoV-2. High-affinity antibodies, generated by previous antigen exposures, have been shown to restrict the specificity and magnitude of secondary B cell responses, and limit recruitment of naïve B cells into the germinal centers[68–70]. Similar to reports after Omicron-based mRNA immunization[27,29,30,71,72], we found that a protein-based bivalent booster vaccine did not elicit considerable de novo B cell response against BA.5 S-specific epitopes but predominantly expanded cross-reactive B cells specific for the epitopes that are shared between ancestral and Omicron strains. Nevertheless, the booster vaccine broadened antibody reactivity to effectively neutralize divergent Omicron sublineages, including the forward drift variant BQ.1.1, whereas neutralization of XBB.1.5 was weaker, in line with previous reports[73]. Although the serum antibodies were highly imprinted by the exposure to ancestral WA-1 proteins, as demonstrated before in mice[70] and men[72], four animals in our study developed some degree of de novo serum IgG reactivity towards BA.5 RBD after XBB.1.5 infection. High copy numbers of viral sgRNA in the BAL were detected in this subgroup, suggesting that high antigen loads and/or repeated Omicron exposures are needed to divert the specificity of the B cell repertoire[70,74]. It must be noted that we used BA.5 Spike and RBD as a proxy for Omicron-derived proteins and did not specifically evaluate the reactivity towards the XBB.1.5 proteins. However, it has been shown that the WA-1 imprinted responses account for majority of the neutralizing capacity against emerging Omicron strains[75,76] and we speculate that de novo reactivity against unique XBB.1.5 epitopes would be similar to what we have shown for BA.5 proteins.

Antibody responses in the upper respiratory tract appear to be highly compartmentalized, and difficult to elicit with intramuscular immunization[3,42,45,77,78]. On the other hand, antibodies in the BAL fluid

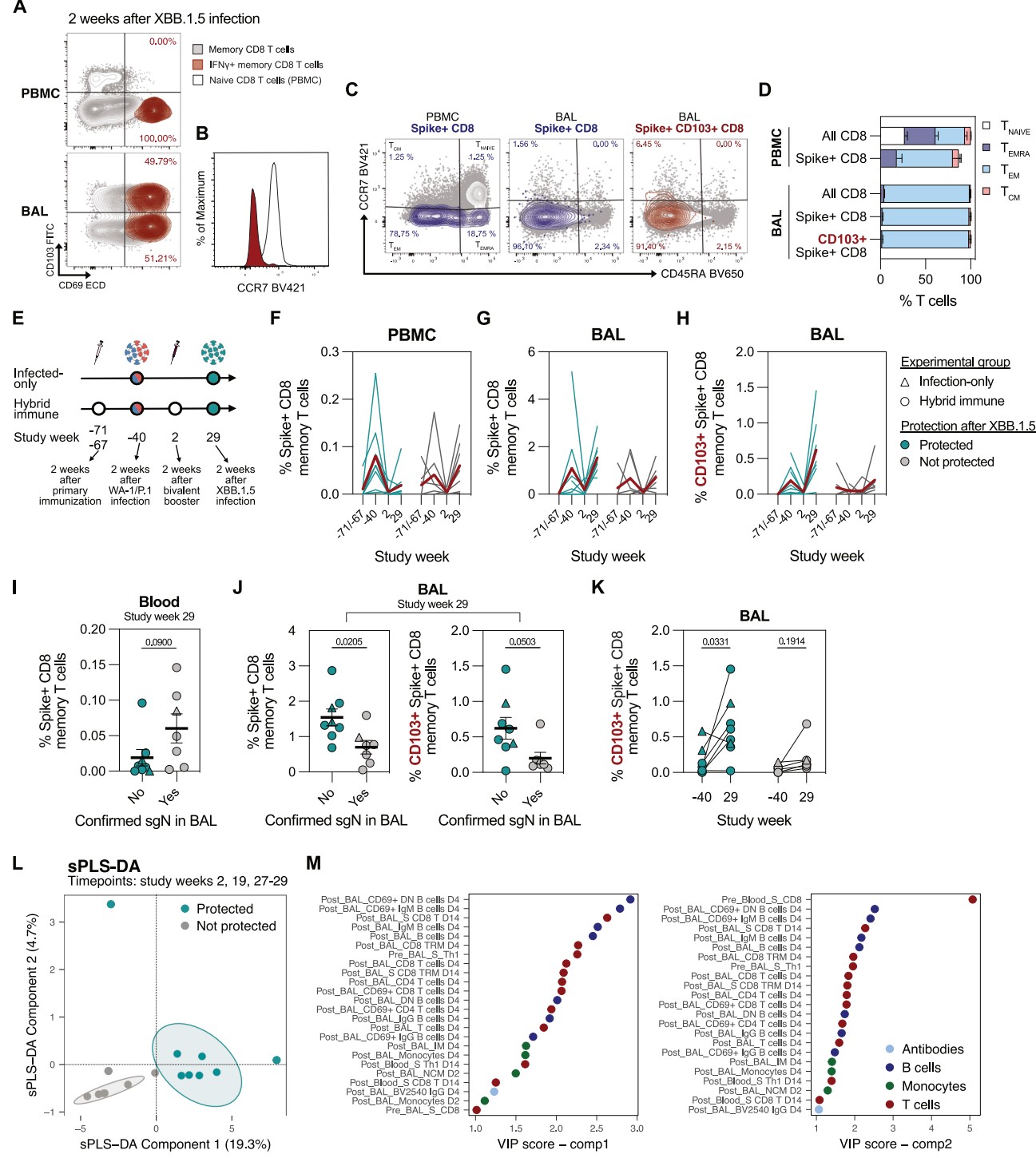

derive both from locally differentiated plasmablasts as well as serum transudates, depending on the participant's exposure history[23,43,79]. We observed a potent increase in WA-1 neutralization in the BAL fluid, but not in the serum of hybrid immune rhesus macaques after XBB.1.5 infection. This indicated a mucosal origin for a proportion of BAL antibodies, which may have been primed by the local antigen encounter during the primary infection[41,43]. Although we were not able to address this, other studies have proposed that antibodies derived from tissue-resident memory B cells may contribute to ameliorating SARS-CoV-2 infection in lower airways and thus disease severity through direct neutralization of the virus as well as Fc-mediated effector functions[80–82].

Whether infection-derived immunity is more efficient at preventing reinfection than hybrid or vaccine-derived immunity remains unclear[12,83,84], especially as the magnitude and localization of CD8 T cell and antibody responses vary based on the exposure history[4,85,86]. In this study, the viral loads in the infected-only group were similar to those in hybrid immune animals. However, there was an extensive time span between the bivalent boost and XBB.1.5 challenge (6 months) as well as between the two consecutive challenges (16 months). Mirroring the general population, S-specific immunity waned over time and reduced the reliance on high antibody levels for protection. Despite the low level of productive infection in both hybrid immune and infected-only group, S- and RBD-binding IgG in the blood increased solely in the

**Fig. 7 | Protection from infection is associated with expansion of Spike-specific tissue-resident CD8 memory T cells and early infiltration of cells into the lungs.** **A** CD69+ CD103+ S-specific tissue-resident CD8 memory T cells ($T_{RM}$) (red) appear in the BAL, but not blood, after XBB.1.5 infection. CD8 memory T cells (gray) are shown for comparison. **B** CCR7 expression on naïve CD8 T cells in blood (white) and S-specific $T_{RM}$ cells in BAL (red). **C** CCR7 and CD45RA surface expression on S-specific CD8 T cells in blood and BAL and S-specific CD8 $T_{RM}$ cells in BAL. **D** Representation of memory T cell phenotypes in blood and BAL in different CD8 T cell populations (n = 15). **E** A schematic timeline of SARS-CoV-2 exposures in hybrid immune and infected-only group. **F−H** Frequency of S-specific CD8 memory T cells in the blood (**F**) and BAL (**G**), and S-specific CD8 $T_{RM}$ cells in the BAL (**H**) at time-points presented in panel (**E**) (n = 3–12 biological replicates per group). **I** Frequency of S-specific CD8 memory T cells in the blood of protected and not protected animals (n = 7–8 biological replicates per group). **J** Frequency of S-specific CD8 memory and S-specific CD8 $T_{RM}$ cells in the BAL of protected and not protected animals (n = 7–8 biological replicates per group). **K** Expansion of S-specific CD8 $T_{RM}$ cells in the lungs of protected animals at SARS-CoV-2 reinfection (week 29) compared to the first infection (week -40) (n = 7–8 biological replicates per group). **L** sPLS-DA analysis, based on immune parameters collected at peak vaccine immunity (week 2), before (week 19), and after XBB.1.5 infection (weeks 27–29). **M** Contribution of different variables to the sPLS-DA components 1 and 2. "Peak_" indicates a measurement two weeks after bivalent booster (study week 2), "Pre_" indicates a measurement before XBB.1.5 infection (study week 19 or 27), and "Post_" indicates a measurement after the XBB.1.5 infection (study week 27 + 2 days to study week 29). Data is presented as arithmetic mean ± SEM (**D, I, J**). Statistical analysis was performed using a non-parametric two-tailed Mann-Whitney test (**I, J**) or non-parametric Kruskal-Wallis test with Dunn's correction for multiple analysis (**K**). Source data are provided as a Source Data file.

infected-only animals. Additionally, we observed a clear segregation of systemic and mucosal T cell responses between the animals with and without active viral replication in the lungs, where protected animals displayed higher T cell responses in the lungs, but not the blood. This may imply that in the animals with productive infection, more viral antigens reached the draining lymph nodes and fueled systemic immune responses[87].

Apart from mucosal antibodies, CD8 $T_{RM}$ cells are instrumental for clearing viral reinfections[35,39,88,89] and accumulate in the tissues following recurrent infections, but not after intramuscular immunization[4,38,90,91]. In our longitudinal dataset spanning several immunizations and two SARS-CoV-2 infections, we observed a significant expansion of S-specific $T_{RM}$ cells after SARS-CoV-2 reinfection in the lungs. Importantly, S-specific $T_{RM}$ cells were absent or very low after the primary immunization and infection or in response to the bivalent booster. Expansion of CD8 $T_{RM}$ cells in response to the second challenge was highly associated with complete suppression of viral replication in the lungs, along with increased influx of monocytes and IgG antibodies into the BAL. In contrast to naïve T cells, which can only be activated by conventional dendritic cells, $T_{RM}$ cells are not restricted and can be re-activated by hematopoietic and non-hematopoietic cells[92]. Antigen presentation by intermediate monocytes in the lung tissue has been shown to stimulate $T_{RM}$ responses upon reinfection[44], thereby promoting tissue-specific immunity. Additionally, CD8 $T_{RM}$ activation triggers exudation of systemic antibodies into mucosal tissue within hours after challenge in mice[93]. In line with this data, higher numbers of activated CD8 $T_{RM}$ cells in the lungs of infected macaques were associated with increased S-specific IgG in the BAL fluid four days after infection. S-specific CD8 $T_{RM}$ cells contracted after infection in all animals, to similar levels detected before the XBB.1.5 infection. It is important to note that we only sampled the BAL compartment, and pulmonary $T_{RM}$ cells can populate both the epithelium as well as the parenchyma[94]. We speculate that at steady state, majority of the $T_{RM}$ cells reside in the lung tissue inaccessible to BAL sampling, leading to underestimation of their numbers. Our data indicates that similar frequencies of $T_{RM}$ cells, observed before XBB.1.5 challenge, could be efficiently re-activated and recruited to the site of infection.

Recent studies of mucosal immunization in mice identified virus-specific CD8 $T_{RM}$ cells as an important immune subset that controls viral infection and prevents onward transmission of respiratory viruses, particularly upon infection with variant strains[88,95,96]. This corroborates our data in hybrid immune NHPs where CD8 $T_{RM}$ cells are associated with a full control of viral replication after heterologous challenge and supports the hypothesis that their role may reach beyond elimination of infected host cells. Several mouse and NHP studies have indicated that a coordinated antibody and CD8 T cell response is required for effective viral control[49,96]. While antibody-targeted epitopes are often under major immune pressure

from antibodies and prone to mutations, T cell responses can be directed against more conserved epitopes. Internal SARS-CoV-2 proteins can also be a target of virus-specific T cell responses, adding another layer of protection[32,85,97], however our analysis focused on the S-specific T cell response due to limited sample availability, prioritizing investigation of immunity elicited after vaccination with S-based vaccines and SARS-CoV-2 infections. Nevertheless, in the face of constant viral evolution, the contribution of T cell and $T_{RM}$ responses is likely substantial, especially when the infection strain is not matched to pre-existing immunity.

In addition to adaptive immune cells, the protection from SARS-CoV-2 reinfection was associated with increased frequency of different monocyte subsets in the BAL early after inoculation of the virus. While excessive activation of the myeloid compartment during acute SARS-CoV-2 and influenza infection has been linked to immunopathology and poor disease outcomes[98,99], a balanced immune response at the site of infection, including the innate immune cells, has been shown to aid in antiviral immune responses[36]. A coordinated immune response at the site of the infection is therefore required for an effective viral clearance, as indicated by the sPLS-DA regression analysis.

Similar to our study, rapid influx of innate immune cells, particularly monocytes, into the upper respiratory tract has been observed during SARS-CoV-2 infection of seronegative adults[63,64], while data about cell recruitment into the lower airways in humans are lacking. Moreover, the serum antibodies are heavily imprinted by the primary exposure in both NHPs and humans[72,100], whereas it is primarily memory B cell targeting conserved regions that expand after vaccination[27,29].

While intramuscular COVID-19 vaccines elicit durable systemic immunity and are effective at preventing severe disease, their capacity to inhibit infection and viral transmission is low[101]. On the other hand, SARS-CoV-2 infection primes strong immune responses in the nasal cavity, lungs and lung-draining lymph nodes[85,102,103], but also carries a significant risk of post-infection sequelae[104,105]. Recently, major efforts have been dedicated to developing COVID-19 mucosal vaccines for induction of virus-specific immune responses in the respiratory tract[38,42,106], which aim to elicit protective immunity at the site of infection. Viral-vectored vaccines appear more suitable for this purpose compared to the mRNA platform, as they mimic viral infection, particularly when delivered intratracheally or in an aerosolized form. These approaches elicited superior antibody and T cell responses in the upper and lower respiratory tract compared to intramuscular immunization and protected NHPs from Omicron challenge[42,45], but the phenotype of the S-specific T cell immunity was not investigated in detail. Future studies should address lung-specific immunity after vaccination and infection, alternative formulation and delivery strategies informing the design of future vaccines for maximum protection at the mucosal surfaces.

Our study highlights the importance of a coordinated immune responses at the site of infection for effective control of heterologous SARS-CoV-2 infection. The protected NHPs displayed increased infiltration of innate and adaptive immune cells into the BAL in the first days after infection, followed by a considerable expansion S-specific lung-resident CD8 $T_{RM}$ cells two weeks after. While both hybrid immunity and infection-derived immunity provided high levels of protection against SARS-CoV-2 challenge, these results underscore the need for the development of mucosal vaccines with the ability to prime tissue-resident immune responses without the risks associated with viral infection.

## Methods

### Vaccine production
For NVX-CoV2373 (WA-1 S), full-length S glycoprotein (GenBank MN908947 nucleotides 21563–25384) was modified by substituting the S1/S2 furin cleavage site from 682-RRAR-685 to 682-QQAQ-685 (3Q). Two proline substitutions were introduced at positions K986P and V987P (2P) to stabilize the full-length SARS-CoV-2 S[107]. The gene sequence was synthetically produced and codon optimized for expression in *Spodoptera frugiperda* (Sf9) cells (GenScript)[108]. For NVX-CoV2540 (BA.5 S), the following substitutions were introduced in addition to the stabilizing 3Q-2P mutations (reference sequence EPI-ISL 12097410.1): T19I, Δ24-26, A27S, Δ69-70, G142D, V213G, G339D, S371F, S373P, S375F, T376A, D405N, R408S, K417N, N440K, L452R, S477N, T478K, E484A, F486V, Q498R, N501Y, Y505H, D614G, H655Y, N679K, P681H, N764K, D796Y, Q954H, and N969K. Matrix-M adjuvant was supplied by Novavax AB (Uppsala, Sweden). Vaccine components (recombinant proteins and Matrix-M) were diluted in the formulation buffer (25 mM sodium phosphate, 300 mM sodium chloride, 0.01% Tween 80, pH 7.2).

### Study design
Animal experiments were conducted following the *Guide for the Care and Use of Laboratory Animals*, NRC 2011, the Swedish Animal Welfare Law and Regulations and the European Directive 2010/63/EU on the protection of animals used for scientific purposes. The study was approved by the Regional animal ethics committee of Stockholm (Stockholms djurförsöksetiska nämnd).

Indian rhesus macaques (*Macaca mulatta*, male and female, 6–8 years old at study start, n = 15) were housed at Astrid Fagraeus Laboratory at Karolinska Institutet (Stockholm, Sweden). Hybrid immune animals (n = 12), generated in a previous study by using Novavax's vaccines and infection with WA-1 or P.1 SARS-CoV-2 virus[46], were immunized with a bivalent protein subunit booster vaccine nine months after earlier SARS-CoV-2 infection. The booster vaccine (0.5 mL) contained 2.5 µg NVX-CoV2373, 2.5 µg NVX-CoV2540, and 50 µg Matrix-M adjuvant and was delivered intramuscularly in the left quadriceps. Animals that have previously only been infected (n = 3), were not immunized but sampled on the same schedule for comparison. Twenty-seven weeks after the bivalent immunization, all animals (hybrid immune and infected-only group) were infected with 8×10^5 PFU of SARS-CoV-2 Omicron XBB.1.5 (isolate hCoV-19/USA/MD-HP40900/2022 (BEI Resources, #NR-59105)). The inoculum was diluted in PBS and delivered intranasally (2 × 0.5 mL) and intratracheally (3 mL). Heparinized blood, serum, bone marrow aspirates, nasal swabs, and BAL were collected regularly throughout the study period as shown in Fig. 1A. Body temperature and body weight were measured at each sampling timepoint.

### Sample processing
Peripheral blood mononuclear cells (PBMCs) were isolated from heparinized blood using a standard gradient density centrifugation protocol with Ficoll-Paque (Cytiva, #17144003). PBMCs were either immediately used for downstream assays or cryopreserved in 10% dimethyl sulfoxide (DMSO, Fisher Scientific, #10206581)/fetal calf serum (FCS, Thermo Fisher, #A5256701). Plasma fraction was cryopreserved neat after isolation. Bone marrow aspirates were collected in blood collection tubes with heparin, processed in the same manner as PBMCs and passed through a 70 µm cell strainer before use.

After collection, BAL samples were centrifuged at 300 × *g* for 10 min to separate the cellular fraction from the BAL fluid. The fluid was concentrated using Amicon Ultra 30 kDa centrifugal units (Merck, #UFC903024) 10-fold before downstream applications. Cellular BAL fraction was passed through a 70 µm cell strainer and used fresh for phenotyping or functional assays.

### ELISA
96-well half-area ELISA plates (Greiner Bio, #738-0032) were coated with recombinant proteins at 1 µg/mL in PBS and incubated at 4 °C overnight. The coating antigens used were vaccine antigens BV2373 (WA-1 S) and BV2540 (BA.5 S) (provided by Novavax), WA-1 RBD (acquired through the Global Health Discovery Collaboratory funded by the Bill & Melinda Gates Foundation) and BA.4/5 RBD (BioTechne, #11229-CV-100). The next day, plates were washed three times with PBS-T (0.05% Tween20 in PBS) and blocked with blocking buffer (5 % (w/v) skimmed milk powder (Oxoid, #16694685) in PBS) for 1 h at room temperature (RT). Plasma or concentrated BAL fluid samples were serially diluted in blocking buffer and added to the plate in duplicates. After 2 hr incubation, the plates were washed three times with PBS-T. Goat anti-monkey IgG-horseradish peroxidase detection antibody (Nordic MUBio, #246-GAMon/IgG(Fc)/PO, 1:15,000 dilution) in blocking buffer was added for 1 h at RT, followed by another PBS-T wash. The development was performed by adding 1-Step Ultra TMB-ELISA substrate (Thermo Fisher, #34028) for 5 min at RT and stopping the reaction with 1 M sulfuric acid. Absorbance was measured at 470 nm with background correction at 570 nm on a Varioskan LUX microplate reader (Thermo Scientific). Data analysis was performed in GraphPad Prism v10.1.0, and $ED_{50}$ (plasma) or endpoint titers (BAL fluid) were calculated using 4-parameter logistic curve fit,

### Competition ELISA
A competition ELISA was performed as above with certain modifications to evaluate the proportion of antibodies binding epitopes exclusive to BA.5 RBD or WA-1 RBD. The plates were coated with BA.4/5 RBD or WA-1 RBD, respectively, and serially diluted plasma samples were preincubated with 20 µg/mL of competitor antigen in blocking buffer (WA-1 RBD or BA.4/5 RBD, respectively) or blocking buffer alone for 30 min at RT. The preincubated samples were transferred into the ELISA plate and incubated for 1.5 h. The detection was performed as described above, and endpoint titers were calculated in GraphPad Prism v10.1.0 using 4-parameter logistic curve fit.

### Pseudoneutralization assays
Pseudoneutralization assays of WA-1 and BA.4/5 (data presented in Fig. 1D, E, Supplementary Figs. 2B, 3C, D and 4G–M) using vesicular stomatitis virus (VSV)ΔG virus with a luciferase reporter and pseudotyped S proteins were performed at Nexelis (Laval, Canada)[109]. WA-1 S pseudoviruses carried wild type WA-1 S protein, and the following mutations were introduced into the BA.5 S pseudoviruses: T19I, L24S, del25-27, del69-70, G142D, V213G, G339D, S371F, S373P, S375F, T376A, D405N, R408S, K417N, N440K, L452R, S477N, T478K, E484A, F486V, Q498R, N501Y, Y505H, D614G, H655Y, N679K, P681H, N764K, D796Y, Q954H, N969K. All samples were heat-inactivated (30 min at 56 °C) before use and assayed in duplicates. Serum was diluted two-fold starting at 1:10 in 96-well plates and incubated with pseudotyped virus particles (approximately 150,000 RLU) for 1 hour to allow binding of serum antibodies. The mixture was then transferred onto a semi-confluent monolayer of Vero E6 cells and incubated overnight at 37 °C and 5% $CO_2$. The next day, luciferase substrate was added to each well

and the plates were read using a luminescence microplate reader with SoftMax Pro GxP software v6.5.1 (Molecular Devices). The serum dilution required to achieve 50% pseudoneutralization ($PNT_{50}$) was calculated using linear regression.

Neutralization of diverse Omicron sublineages (data presented in Fig. 1F–H, Supplementary Fig. 2C and Fig. 3E, F) was evaluated in a pseudotyped lentiviral system[110]. Each newly produced lot of pseudovirus was titrated under assay conditions to determine the working dilution of target RLU of 100,000 prior to testing. The assay was performed using a HEK293T cell line stably expressing hACE2 (HEK293T/ACE2 obtained from Creative Biogene). Heat-inactivated serum samples (56 °C for 30 min) were serially diluted three-fold in reduced serum Opti-MEM medium (Gibco, #31985070) starting at a 1:50 dilution in a 96-well plate. Stock solution of SARS-CoV-2 pseudoviruses, corresponding to 100,000 RLU, was preincubated with the serum samples at 37 °C for 1 h. $2 \times 10^4$ HEK293T/hACE2 cells in HEK293T cell culture medium (DMEM without phenol red + 5% FBS + 1% Penicillin/streptomycin/glutamine) with 1.25 μg/mL puromycin were added to the wells, followed by incubation for 72 h at 37 °C. After incubation, 50 μL BrightGlo Luciferase Substrate (Promega, #E2620) was added to each well. Plates were incubated for 5 min at room temperature without ambient light. Viral entry into the cells was determined by measuring the luminescence with a SpectraMax iD3 microplate reader (Molecular Devices). Pseudovirus neutralizing antibody titer of the serum was determined through the absence or reduction of luminescence in a well. Data were analyzed and neutralization curves were generated in GraphPad Prism for each sample; 50% pseudovirus neutralization titers ($NT_{50}$) were calculated using 4-parameter curve fitting. No-serum wells were present on each plate along with at least one positive and negative monoclonal antibody for each pseudovirus tested.

## Memory B cell flow cytometry

Recombinant proteins (BV2373 (WA-1 S), BV2540 (BA.5 S), and WA-1 RBD) were biotinylated using EZ-Link Micro Sulfo-NHS-LC Biotinylation Kit (Thermo Fisher, #21935) according to the manufacturer's instructions. Protein probes were created by coupling biotinylated proteins to streptavidin (SA)-fluorophore complexes (SA-APC, SA-PE and SA-BV421, all Biolegend, #405243, #405245, and #405225). PBMC samples were stained with 100 ng of respective protein probes for 20 min at 4 °C, followed by a master mix containing 7-aminoactinomycin D (7-AAD, Thermo Fisher, #A1310) and antibodies listed in Supplementary Table 2 for 20 min at 4 °C. The samples were washed with PBS with 2% heat-inactivated FCS and fixed with 1% formaldehyde solution. Acquisition was performed using BD LSRFortessa cell analyzer, and the data was analyzed using FlowJo software v.10.7.1 (FlowJo).

## Plasma cell ELISpot

Antigen-specific antibody-secreting cells in the bone marrow were measured by ELISpot[46]. Multiscreen IP filter 96-well ELISpot plates (Millipore, #MSIPS4510) were activated with 35% ethanol for 30 seconds and washed with PBS. The plates were coated with Affinity Pure goat anti-human IgG Fc fragment-specific antibody (Jackson ImmunoResearch, #109-005-008) at 1 μg/mL in PBS and incubated at 4 °C overnight. The next day, the plates were washed with PBS and blocked using complete medium (RPMI 1640, 10 % heat-inactivated FCS, 100 U/mL penicillin, 100 mg/mL streptomycin, 2 mM L-glutamine). Bone marrow mononuclear cells were serially diluted, plated in duplicates and incubated overnight at 37 °C and 5% $CO_2$. Following a wash with PBS-T, biotinylated protein probes were added for 1.5 h at 37 °C (0.25 μg/mL goat anti-human IgG Fc fragment-specific antibody (Jackson ImmunoResearch, #109-065-008), 1 μg/mL WA-1 S/BA.5 S/WA-1 RBD/ovalbumin (OVA, Invivogen, #vac-stova)). The plates were then washed with PBS-T, and streptavidin-conjugated alkaline phosphatase (Mabtech, #3310-10) was added at 1:1000 dilution for 30 min

at 37 °C. After another PBS-T wash, the development was performed by adding nitro blue tetrazolium 5-bromo-4-chloro-3' indolyphosphate substrate (Mabtech, #3650-10) for 7 min at RT. The spots were counted using an AID ELISpot reader (Autoimmun Diagnostika). Background reactivity was subtracted based on OVA wells.

## RNA extraction and viral load quantification by RT-qPCR

At indicated timepoints after XBB.1.5 infection, samples from the lower and upper respiratory tract were collected for evaluation of viral load. 1 mL BAL fluid was thoroughly mixed with 1 mL RNAzol BD (Molecular Research Center, #MR-RB192-100) and 10 μL acetic acid (Sigma-Aldrich, #A6283), and immediately frozen using dry ice. Flocked nasal swabs (Copan, #CP503CS01) were collected, submerged into 1 mL PBS containing 1 μL SUPERase-In RNase Inhibitor (Invitrogen, #AM2694) and immediately frozen using dry ice. RNA extraction was performed with RNAzol BD column kit (Molecular Research Center, #MR-RC292) according to manufacturer's instructions.

TaqMan Fast Virus 1-Step Master Mix (Applied Biosystems, #4444432) was used for the RT-qPCR reactions measuring copies of subgenomic N gene transcripts (sgN), genomic N gene transcripts (gN) and RNAseP transcripts. Primer and probe sequences are listed in Supplementary Table 3. In vitro transcribed RNA standards for sgN and gN were produced using MEGAScript T7 Transcription kit (Invitrogen, #AM1333)[111]. RT-qPCR reactions were run in 384-well plates on a CFX384 Touch Real-Time PCR Detection System with CFX Manager software (v3.1, BioRad) using the following program: 5 min at 50 °C, 20 s at 95 °C, and 40 cycles of 15 s at 95 °C and 60 s at 60 °C. For sgN and gN, the lower level of detection was 150-180 copies per reaction. For RNAseP, samples with a Ct value below 40 were assumed positive.

## T cell recall assay

To measure S-specific T cell responses in blood and BAL, PBMCs and cellular BAL fraction were cultured overnight at 37 °C and 5 % $CO_2$ in 96-well U-bottom plates in the presence of WA-1 S overlapping peptides, resuspended in DMSO (2 μg/mL, JPT Peptide Technologies, #PM-WCPV-S-1) and Brefeldin A (10 μg/mL, Invitrogen, #B7450). An equivalent concentration of DMSO was used as negative control for each sample. The next day, the cells were stained with Live/Dead Blue dye (Invitrogen, #L34962) for 5 min at 4 °C, followed by surface staining for 20 min at 4 °C with the antibodies listed in Supplementary Table 2. The samples were fixed and permeabilized with the Cytofix/Cytoperm kit (BD, #554722) according to manufacturer's instructions, and stained with the intracellular staining panel listed in Supplementary Table 2 for 20 min at 4 °C. The samples were washed and fixed with 1 % formaldehyde solution before acquisition on BD LSRFortessa cell analyzer. The data were analyzed with FlowJo software v.10.7.1 (FlowJo). Background reactivity was calculated by subtracting frequencies of CD69+ cytokine+ cells in DMSO-stimulated wells from frequencies of CD69+ cytokine+ cells in peptide-stimulated wells.

## BAL immunoprofiling

Freshly isolated cellular BAL fractions collected at steady state, two and four days after XBB.1.5 infection were phenotyped using flow cytometry. Prior to staining, the cells were washed with PBS supplemented with 2% FCS. The samples were then stained with Live/Dead Blue dye (Invitrogen, #L34962) for 5 min at 4 °C, followed by a panel of antibodies listed in Supplementary Table 2 for 20 min at 4 °C. The cells were washed again prior to fixation with 1% PFA. Data acquisition was performed on BD LSRFortessa cell analyzer, and data was analyzed with FlowJo software v.10.7.1 (FlowJo).

## sPLS-DA analysis

sPLS-DA analysis was chosen because it can define key variables that explain the separation between different groups, can handle large

amounts of data from small sample numbers and is not affected by colinearity. sPLS-DA analysis was performed in RStudio (R version 4.2.1) using the package mixOmics v6.22.0[112]. Immune variables, measured at different timepoints, were used as the input, and the samples were classified based on the status of the NHP after XBB.1.5 infection (i.e. protected versus not protected).

## Statistical analyses

Unless mentioned otherwise, statistical analyses were performed in GraphPad Prism v.10.1.0. Due to small sample sizes, non-parametric tests were used. All statistical tests were two-sided. Mann-Whitney test or Wilcoxon signed-rank test were used for comparison of two unpaired and paired group, respectively. When comparing three or more groups, Kruskal–Wallis test was performed for unpaired and the Friedman test for paired samples. If multiple comparisons were performed at once, Dunn's correction for multiple comparisons was also applied. Correlations were evaluated with Spearman correlation analysis.

## Reporting summary

Further information on research design is available in the Nature Portfolio Reporting Summary linked to this article.

## Data availability

All data are included in the Supplementary Information or available from the authors, as are unique reagents used in this Article. The raw numbers for charts and graphs are available in the Source Data file whenever possible. Source data are provided with this paper.

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

## Acknowledgements

The authors wish to thank the personnel at the Astrid Fagraeus laboratory at Karolinska Institutet for expert assistance and care of the NHPs. The following reagent was obtained through BEI Resources, NIAID, NIH: SARS-Related Coronavirus 2, Isolate hCoV-19/USA/MD-HP40900/2022 (Lineage XBB.1.5; Omicron Variant) (WCCM), NR-59105, contributed by Dr. Andrew S. Pekosz. This work was funded by The Bill & Melinda Gates Foundation grant INV-019352 (to K.Lo.), Swedish Research Council grants 2019-01036 and 2020-05829 (to K.Lo.), Knut and Alice Wallenberg Foundation through SciLifeLab and Karolinska Institutet grant VC-2021-0017 (to K.Lo.), and intramural faculty salary grants from Karolinska Institutet (to K.Le.).

## Author contributions

Conceptualization: K.Le., M.G.-X., N.P., G.S., K.Lo. Formal analysis: K.Le., J.F.T. Funding acquisition: K.Lo. Investigation: K.Le., H.F., M.M., L.P.V., B.E. Methodology: K.Le., H.F., J.F.T. Resources: M.G.-X., N.P., G.S. Supervision: G.M., K.Lo. Visualization: K.Le. Writing – original draft: K.Le., K.Lo. Writing – review & editing: all authors.

## Funding

## Competing interests

M.G.-X., N.P., J.F.T., and G.S. are employees of Novavax Inc. The remaining authors declare no competing interests.
