## [Transparent Peer Review file · Nature Communications]

Coordinated early immune response in the lungs is required for effective control of SARS-CoV-2 replication

Corresponding Author: Professor Karin Lore

Version 0:

Reviewer comments:

Reviewer #1

(Remarks to the Author)

Lenart et al. studied the immune mechanisms contributing to the protection against COVID-19 through the rhesus macaque model. These rhesus macaques were immunized (or not), challenged with SARS-CoV-2 WA.1/P.1, and re-challenged with XBB.1.5. The induced immune responses were comprehensively analyzed. They found that early immune activation of both innate and adaptive immune cells in the respiratory tract conferred protection. This study is interesting and well-performed. My suggestions are listed as follows.

Major:

1. A table describing the details of every rhesus macaque is suggested. Because the two-dose group animals were vaccinated with 2x WA-1 Spike or 2x P.1 Spike and the first challenge experiment was conducted with WA-1 or P.1, a summarized table can give specific and detailed information on what these animals received.
2. Figure 2. The hybrid immune group showed higher viral replication in the respiratory tract after the XBB.1.5 challenge than the infection-only group. The explanation for this result needs to be discussed.
3. These animals were infected with WA-1/p.1 and reinfected with XBB.1.5. How is the immune imprinting after twice infection with SARS-CoV-2? Did the infection of XBB.1.5 elicit the de novo XBB.1.5 responses?
4. The responses/symptoms after COVID-19 vaccine immunization and SARS-CoV-2 infection in humans are heterogeneous. The authors studied the protected group and not protected group of animals after the XBB.1.5 challenge. Compare these results with the other findings from humans and dedicates the potential guidance for new COVID-19 vaccine or drug development.

Minor:

1. The Abstract did not indicate the variant or strain these animals first challenged.
2. The statistical differences are suggested to be added in Figures 4 and S5.
3. Lines 194-202. The referenced figures (Fig.S4) are incorrect.

Reviewer #2

(Remarks to the Author)

Lenart et al present a cohesive set of results outlining the importance of early cellular responses in the lungs to protective responses against SARS-CoV-2 infection. The findings on T cell importance, though not proven to be causative, are compelling and make this paper a good fit to be published in Nature Communications, there are few such good pieces of evidence directly measuring T_{rm} responses in such NHP challenge studies the literature.

The competition ELISA most likely should have been done in a sandwich ELISA format or a flow-cytometry format to avoid the confounding affects of coating antigen deformation which comes with direct ELISAs. The assay should have at least been done in both directions (coating with BA.5 and competing with WT as well as coating with WT and competing with BA.5 RBD). I would strongly recommend this assay to be repeated in a more controlled manner to clarify that what is being reported is actually what is being measured.

Though mentioned in the methods, the small sample size, and the impact that may have on the robustness of the results, should also be discussed in the results and discussion of the paper.

I look forward to a revised version of this manuscript and the subsequent publication

Reviewer #3

(Remarks to the Author)

This manuscript demonstrates that hybrid immune rhesus macaques boosted with Novavax's SARS-CoV-2 bivalent WA-1/BA.5 vaccine developed strong cross-reactive S-specific immune responses. Upon reinfection with Omicron XBB.1.5, viral replication remained low, indicating substantial protection. The study highlights that this protection was driven by early activation of innate and adaptive immune cells, including monocytes, S-specific IgG, and lung tissue-resident CD8 T cells in the respiratory tract. This paper presents significant findings that contribute to our understanding of SARS-CoV-2 immunity using a non-human primate model. I believe it has strong merit for publication in Nature Communications. However, to further enhance the clarity and depth of the study, I request that the authors address the following questions before acceptance:

1) lines 162–170,

The study demonstrates the expansion of cross-reactive WA-1/BA.5+ B cells in hybrid immune animals, while de novo BA.5-specific responses were limited. Have the specific epitopes driving this cross-reactivity been identified?

2) lines 173–156,

The authors observed a stronger immune influx in the BAL following reinfection compared to vaccination alone. Could they clarify the mechanisms underlying this difference? Additionally, would extending the observation period after vaccination potentially reveal a similar recruitment of immune cells to the lungs?

3) lines 241–249,

Could the authors elaborate on the key factors driving the rapid expansion of lung TRM cells upon reinfection? Additionally, how might alternative vaccination strategies, such as mucosal delivery, enhance TRM induction and improve protection against highly transmissible variants?"

What is the duration of persistence for lung TRM cells following infection or vaccination? Additionally, are these cells capable of conferring protection against reinfection over long-term periods, such as months or years?

4) line 259 – 261,

How well do the immune mechanisms observed in rhesus macaques, particularly those involving lung-resident CD8 T cells and monocyte responses, translate to humans? Additionally, could clinical studies employing mucosal sampling methods, such as BAL or nasal swabs, help validate the role of early immune activation in human SARS-CoV-2 reinfections?

5) line 372 – 374,

Should future vaccine designs prioritize the stimulation of mucosal immunity, such as through nasal vaccines, to enhance respiratory tract responses? Furthermore, how can booster vaccination strategies be optimized to sustain both systemic and local immune responses over an extended period?

Version 1:

Reviewer comments:

Reviewer #1

(Remarks to the Author)

The authors have addressed the issues. I have no more concerns.

Reviewer #2

(Remarks to the Author)

Reviewer #3

(Remarks to the Author)

The authors have thoroughly addressed all of my previous comments and questions. Their responses were scientifically sound and provided appropriate clarification or expansion of key points related to cross-reactive B cell responses, lung immune cell dynamics, tissue-resident memory T cell (TRM) persistence, and the relevance of their findings to human immunity and vaccine strategies.

The revised manuscript is now substantially improved, particularly in the Discussion section where important mechanistic

insights and translational relevance are now more clearly articulated. The additional data on TRM longevity and the extended discussion of mucosal vaccination approaches add meaningful depth to the manuscript.

Given these revisions, I find the manuscript to be scientifically robust, clearly written, and suitable for publication in Nature Communications.

Reviewer #1 (Remarks to the Author):

Lenart et al. studied the immune mechanisms contributing to the protection against COVID-19 through the rhesus macaque model. These rhesus macaques were immunized (or not), challenged with SARS-CoV-2 WA.1/P.1, and re-challenged with XBB.1.5. The induced immune responses were comprehensively analyzed. They found that early immune activation of both innate and adaptive immune cells in the respiratory tract conferred protection. This study is interesting and well-performed.

We thank the reviewer for the supportive comments about our study.

My suggestions are listed as follows.

Major:

1. A table describing the details of every rhesus macaque is suggested. Because the two-dose group animals were vaccinated with 2x WA-1 Spike or 2x P.1 Spike and the first challenge experiment was conducted with WA-1 or P.1, a summarized table can give specific and detailed information on what these animals received.

The manuscript now includes a Table S1 that provides detailed exposure history for each rhesus macaque included in this study.

2. Figure 2. The hybrid immune group showed higher viral replication in the respiratory tract after the XBB.1.5 challenge than the infection-only group. The explanation for this result needs to be discussed.

This is a relevant point raised by the reviewer. However, this study was not powered to assess differences in protective effects of the vaccination or lack thereof against SARS-CoV-2 infection. Figure 2B, which displays the peak viral loads in the upper or lower respiratory tract during the 14-day follow-up, indeed appears to show elevated viral loads in the hybrid immune group. This is due to a significant limitation that one of the NHPs in the infected-only group could not be sampled on day 2 after challenge, based on the veterinarian's advice, as a small bleed was observed in the trachea. This bleed resolved spontaneously by day 4. We therefore lack a positive viral load reading from this particular NHP, although immunologically it is clear the NHP was infected with SARS-CoV-2 based on elevated antibody titers in the blood and BAL, which follow the same kinetics as other infected-only NHPs.

To further explore the differences in the viral load between groups, we added a plot showing area under curve (AUC) measurements of the sgN viral loads for each NHP after XBB.1.5 infection, now shown in Fig. S3A. The AUC values represent both the peak of the viral load as well as the duration, meaning that the NHPs that sustained higher viral loads for longer have higher AUC values. While the difference between groups remains non-significant and there is a large spread in the data, the infected-only animals display some of the highest AUC values, particularly in the upper respiratory tract. We speculate that had we collected day 2 post-infection samples for the particular NHP, the peak viral loads between the groups would be even more comparable. This result is now more extensively discussed in the results section, lines 122-126 and in the discussion, lines 306-315.

3. These animals were infected with WA-1/P.1 and reinfected with XBB.1.5. How is the immune imprinting after twice infection with SARS-CoV-2? Did the infection of XBB.1.5 elicit the de novo XBB.1.5 responses?

In this study, the focus was on the immunological analyses against the Omicron BA.5 strain (included in the bivalent booster) as a proxy for a divergent Omicron Spike antigens due to limited sample volumes. While BA.5 Spike and WA-1 Spike differ by 31 amino acids, BA.5

Spike and XBB.1.5 Spike differ by 16 amino acids, 9 of which are located in the RBD (Rössler *et al*, PMID: 39754717).

Our profiling of the memory B cell and antibody repertoire shows that the B cell responses are heavily imprinted by the primary exposure (in line with Schiepers *et al*, PMID: 36646114 and Tortorici *et al*, PMID: 38490197), and published literature shows that the imprinted responses account for majority of the neutralizing capacity even against emerging Omicron strains (Rössler *et al*, PMID: 39754717; Zou *et al*, PMID: 36734885). Given a low frequency of antibodies recognizing epitopes that are unique to BA.5 RBD even after two exposures, we speculate that the *de novo* XBB.1.5 response is similarly limited after a single XBB.1.5 infection. This has now been included in the discussion, lines 331-335.

4. The responses/symptoms after COVID-19 vaccine immunization and SARS-CoV-2 infection in humans are heterogeneous. The authors studied the protected group and not protected group of animals after the XBB.1.5 challenge. Compare these results with the other findings from humans and dedicates the potential guidance for new COVID-19 vaccine or drug development.

Similar to the human population, the immune response to COVID-19 mRNA or protein vaccination in NHPs are somewhat heterogeneous, although almost all individuals respond to vaccination with increased antibody titers compared to the pre-vaccination samples (Muecksch *et al*, PMID: 35447027; Vogel *et al*, PMID: 33524990; Turner *et al*, PMID: 34182569). On the other hand, the clinical presentation to COVID-19 infection in humans can vary from asymptomatic to severe or fatal disease, while the COVID-19 infection in NHPs is predominantly mild (Nelson *et al*, PMID: 35271298). Nevertheless, both human and NHP studies report altered tropism between ancestral and Omicron strains, the former favoring the lower respiratory tract, while the latter primarily infects the upper airways (Meng *et al*, PMID: 35104837; Gagne *et al*, PMID: 35447072). Moreover, a recent study demonstrated an influx of different monocyte subsets into the upper and lower airways during acute SARS-CoV-2 infection with the ancestral WA-1 strain (Österberg *et al*, PMID: 40041475), which aligns with our findings of the influx of monocytes into the lower airways early after XBB.1.5 infection. This highlights the similarities between the properties of the viral infection and immune responses across species. The expanded comparison between NHP and human studies is now included in the discussion, lines 306-315 and 401-405.

Regarding future vaccine development, our data as well as other NHP studies (McMahan *et al*, PMID: 38096903; Gagne *et al*, PMID: 39227514) support development of formulation and delivery strategies for mucosal vaccination, aiming to induce localized protective immunity in the respiratory tract. The section in the discussion about future guidance has now been refined, lines 416-419.

Minor:

1. The Abstract did not indicate the variant or strain these animals first challenged. Information about the animals' earlier exposures to SARS-CoV-2 was added to the abstract.
2. The statistical differences are suggested to be added in Figures 4 and S5. Statistical analyses were added to panels 4C, 4F, S5C and S5F.
3. Lines 194-202. The referenced figures (Fig.S4) are incorrect.

The references to supplementary figures related to the T cell assay have been updated to refer to Figure S6.

Reviewer #2 (Remarks to the Author):

Lenart et al present a cohesive set of results outlining the importance of early cellular responses in the lungs to protective responses against SARS-CoV-2 infection. The findings on T cell importance, though not proven to be causative, are compelling and make this paper a good fit to be published in Nature Communications, there are few such good pieces of evidence directly measuring Tm responses in such NHP challenge studies the literature.

We thank the reviewer for the encouraging comments about our study.

The competition ELISA most likely should have been done in a sandwich ELISA format or a flow-cytometry format to avoid the confounding affects of coating antigen deformation which comes with direct ELISAs. The assay should have at least been done in both directions (coating with BA.5 and competing with WT as well as coating with WT and competing with BA.5 RBD). I would strongly recommend this assay to be repeated in a more controlled manner to clarify that what is being reported is actually what is being measured.

We thank the reviewer for constructive criticism. We have now repeated the ELISA in a reverse order as suggested. This approach also demonstrated a striking imprinting of the antibody response by the prime WA-1 antigen, confirming the previous data. Moreover, the same phenomenon of imprinting was recently described both in mice and humans (Schiepers *et al*, PMID: 36646114; Tortorici *et al*, PMID: 38490197). The new data is included in the manuscript in Fig. S4C-D. These results are described in lines 169-179 and discussed in lines 326-327.

The competition ELISA including direct coating of antigen is a robust assay that has been performed in a similar manner by a number of labs and for different antigens (RSV: Chang *et al*, PMID: 36542692; Ols *et al*, PMID: 37689061 and SARS-CoV-2: Carreño *et al*, PMID: 37141905; Pušnik *et al*, PMID: 38600072; Carreño *et al*, PMID: 40042313). In addition, RBD is biophysically a very stable protein (Ellis *et al*, PMID: 34267764; Moro-Pérez *et al*, PMID: 37150832), and direct coating of ELISA antigens has been widely used in commercial (<https://www.thermofisher.com/elisa/product/Human-SARS-CoV-2-Spike-Trimer-IgG-ELISA-Kit/BMS2325>, <https://www.acrobiosystems.com/P3259-SARS-CoV-2-Spike-protein-RBD-Coated-Plates-Clear-96-Well.html>) as well as in in-house assays (Amanat *et al*, PMID: 32398876; Cho *et al*, PMID: 34619745; Walls *et al*, PMID: 34619077), with no reports of conformational instability known to us.

We therefore believe that our ELISA results, also corroborated by the reports of other groups, are robust and describe the lack of a *de novo* antibody response against the epitopes unique to BA.5 RBD.

Though mentioned in the methods, the small sample size, and the impact that may have on the robustness of the results, should also be discussed in the results and discussion of the paper.

The impact of the small sample sizes on the results have now been further emphasized in the discussion, lines 298-299.

I look forward to a revised version of this manuscript and the subsequent publication.

Reviewer #3 (Remarks to the Author):

This manuscript demonstrates that hybrid immune rhesus macaques boosted with Novavax's SARS-CoV-2 bivalent WA-1/BA.5 vaccine developed strong cross-reactive S-specific immune responses. Upon reinfection with Omicron XBB.1.5, viral replication remained low, indicating substantial protection. The study highlights that this protection was driven by early activation of innate and adaptive immune cells, including monocytes, S-

specific IgG, and lung tissue-resident CD8 T cells in the respiratory tract. This paper presents significant findings that contribute to our understanding of SARS-CoV-2 immunity using a non-human primate model. I believe it has strong merit for publication in Nature Communications. However, to further enhance the clarity and depth of the study, I request that the authors address the following questions before acceptance:

We thank the reviewer for the kind comments about our study.

1) lines 162–170,

The study demonstrates the expansion of cross-reactive WA-1/BA.5+ B cells in hybrid immune animals, while de novo BA.5-specific responses were limited. Have the specific epitopes driving this cross-reactivity been identified?

We agree with the reviewer that identifying specific cross-reactive epitopes driving the expansion of memory B cells would be an interesting avenue of investigation to pursue, however it is outside the scope of this study. The same question was addressed in earlier studies, showing with cryoEM structures of RBD:Fab complexes that different epitopes in the hACE2-binding region on RBD can drive the evolution of broadly-neutralizing antibodies (Windsor *et al*, PMID: 35536154).

2) lines 173–156,

The authors observed a stronger immune influx in the BAL following reinfection compared to vaccination alone. Could they clarify the mechanisms underlying this difference? Additionally, would extending the observation period after vaccination potentially reveal a similar recruitment of immune cells to the lungs?

Increased influx of immune cells into the lungs after infection compared to vaccination is most likely a consequence of the intranasal and intratracheal administration of the viral inoculum and the subsequent pulmonary inflammation, in contrast to intramuscular vaccination (Mettelman, Allen and Thomas, PMID: 35545027). While we did not directly measure the composition of the immune cells in the BAL fluid after vaccination due to ethical limitations in sampling frequency and volume, it is unlikely that intramuscular vaccination would promote migration of immune cells into the respiratory tract at any timepoint, as we have previously shown that intravenous, but not subcutaneous vaccine delivery is required for efficient targeting and activation of lung T cells (Thompson *et al*, PMID: 29769448). Furthermore, another NHP study demonstrated that in contrast to intradermal vaccination, intravenous delivery of the BCG vaccine induces prolonged activation and migration of immune cells into the lungs (Peters *et al*, PMID: 39742484), however that is likely due to both intravenous delivery and BCG being a replicating, live attenuated vaccine.

3) lines 241–249,

Could the authors elaborate on the key factors driving the rapid expansion of lung TRM cells upon reinfection? Additionally, how might alternative vaccination strategies, such as mucosal delivery, enhance TRM induction and improve protection against highly transmissible variants?"

We thank the reviewer for this very relevant question. In the revised manuscript, the section on potential mechanisms driving the expansion of lung T_{RM}s has been further expanded. Once T_{RM}s are induced by local antigen encounter (Gebhardt *et al*, PMID: 19305395; Devarajan *et al*, PMID: 37776519; Zens *et al*, PMID: 27468427), it has been proposed that they are not restricted by conventional dendritic cells and can be reactivated by different hematopoietic and non-hematopoietic cells (Low *et al*, PMID: 32525985), with a particular role for inflammatory (CD14+CD16+) monocytes (Dunbar *et al*, PMID: 31723250). This has been added to the discussion, lines 366-368.

In line with previous reports, our data indicates that unlike prior infection, repetitive intramuscular vaccination does not induce a lung T_{RM} population. Taken together with recent studies of mucosal vaccination (McMahan *et al*, PMID: 38096903; Gagne *et al*, PMID: 39227514), the most effective way to generate robust protective responses in the respiratory tract is likely through optimizing mucosal vaccine delivery. This has been further emphasized in the updated section about future guidance in the discussion, lines 416-419.

What is the duration of persistence for lung T_{RM} cells following infection or vaccination? Additionally, are these cells capable of conferring protection against reinfection over long-term periods, such as months or years?

Studies in organ donors and transplant recipients demonstrate that T_{RM} populations can persist for extended periods of time (Miron *et al*, PMID: 34127056; Snyder *et al*, PMID: 30850393), although studies in mice suggest that antigenic restimulation is required for T_{RM} survival (Uddbäck *et al*, PMID: 32518368). In our study, the T_{RM}s were able to respond 6 months after last vaccination and 15 months after last infection.

We measured the frequency of Spike-specific CD103⁺ CD8⁺ T cells in the lungs at study week 57 (30 weeks after the XBB.1.5 infection). The data is now included in the manuscript as Fig. S8K-M and described in lines 270-273 and 373-379. The frequency of Spike-specific T_{RM} cells at week 57 was similar to the pre-challenge frequency at week 19, which is a clear reduction from the peak post-infection timepoint. As our study utilized non-terminal sampling procedures for longitudinal sampling of the respiratory tract, we cannot rule out the possibility that the relevant T_{RM} populations reside deeper within the parenchyma rather than close to alveoli. This could result in increased detection of T_{RM} cells during infection when the tissues are more permeable, potentially skewing the results.

4) line 259–261,

How well do the immune mechanisms observed in rhesus macaques, particularly those involving lung-resident CD8 T cells and monocyte responses, translate to humans? Additionally, could clinical studies employing mucosal sampling methods, such as BAL or nasal swabs, help validate the role of early immune activation in human SARS-CoV-2 reinfections?

This is an excellent question. Both species are highly similar in their anatomy, physiology and immune system (Estes *et al*, PMID: 29556017) and share a similar distribution of pulmonary T_{RM} subsets (Humphries *et al*, PMID: 36845117). Due to their similarity, majority of assays and reagents can be used interchangeably, and most immune subsets are defined by the same surface receptor expression patterns. This makes most findings in the NHP models highly translatable to the human population.

While sampling of different respiratory compartments in humans is feasible (Baharom *et al*, PMID: 27183618; Falck-Jones *et al*, PMID: 33492309; Ramirez *et al*, PMID: 39085605), it is also relatively uncomfortable for the study subjects and rarely performed longitudinally (in particular BAL sampling). As noted in our response to Reviewer #1 above we have now expanded the discussion on comparisons between NHP and human studies (lines 306-315 and 401-405).

5) line 372–374,

Should future vaccine designs prioritize the stimulation of mucosal immunity, such as through nasal vaccines, to enhance respiratory tract responses? Furthermore, how can booster vaccination strategies be optimized to sustain both systemic and local immune responses over an extended period?

We believe that exploring mucosal delivery is one of the plausible strategies to improve vaccine-induced immunity against respiratory viruses. For example, one mouse study suggested that combining subcutaneous and intranasal vaccination is superior to intranasal vaccination alone in inducing lung-specific immune responses (Uddback *et al*, PMID: 26831578). Moreover, NHP studies showed that boosting with adenoviral vectored-vaccines and intratracheal or aerosolized delivery confers better protection from infection than conventional vaccine delivery (McMahan *et al*, PMID: 38096903; Gagne *et al*, PMID: 39227514). A recent phase I clinical study of an intranasal SARS-CoV-2 vaccine failed to meet its primary endpoint to demonstrate efficacy (Zhu *et al*, PMID: 37979588). Notably, their vaccine performed worse in people with pre-existing immunity, underscoring the need for improved formulation and delivery especially in a boost setting.

We would like to again thank the reviewers for providing thorough and constructive criticism that has helped us to improve our manuscript.